# Estimation of gas record alteration in very low accumulation ice cores

Kévin Fourteau[1], Patricia Martinerie[1], Xavier Faïn[1], Alexey A. Ekaykin[2], Jérôme Chappellaz[1], and Vladimir Lipenkov[2]

[1]Univ. Grenoble Alpes, CNRS, IRD, Grenoble INP, IGE, F-38000 Grenoble, France
[2]Climate and Environmental Research Laboratory, Arctic and Antarctic Research Institute, St. Petersburg, 199397, Russia

**Correspondence:** kevin.fourteau@univ-grenoble-alpes.fr or patricia.martinerie@univ-grenoble-alpes.fr

**Abstract.** We measured the methane mixing ratios of enclosed air in five ice core sections drilled on the East Antarctic Plateau. Our work aims to study two effects that alter the recorded gas concentrations in ice cores: layered gas trapping artifacts and firn smoothing. Layered gas trapping artifacts are due to the heterogeneous nature of polar firn, where some strata might close early and trap abnormally old gases that appear as spurious values during measurements. The smoothing is due to the combined

effects of diffusive mixing in the firn and the progressive closure of bubbles at the bottom of the firn. Consequently, the gases trapped in a given ice layer span a distribution of ages. This means that the gas concentration in an ice layer is the average value over a certain period of time, which removes the fast variability from the record. Here, we focus on the study of East Antarctic Plateau ice cores, as these low accumulation ice cores are particularly affected by both layering and smoothing. We use high-resolution methane data to test a simple trapping model reproducing the layered gas trapping artifacts for different accumulation

conditions typical of the East Antarctic Plateau. We also use the high-resolution methane measurements to estimate the gas age distributions of the enclosed air in the five newly measured ice core sections. It appears that for accumulations below $2\,\mathrm{cm\,ice\,equivalent\,yr^{-1}}$ the gas records experience nearly the same degree of smoothing. We therefore propose to use a single gas age distribution to represent the firn smoothing observed in the glacial ice cores of the East Antarctic Plateau. Finally, we used the layered gas trapping model and the estimation of glacial firn smoothing to quantify their potential impacts on a

hypothetical million-and-a-half years old ice core from the East Antarctic Plateau. Our results indicate that layering artifacts are no longer individually resolved in the case of very thinned ice near the bedrock. They nonetheless contribute to slight biases of the measured signal (less than $10\,\mathrm{ppbv}$ and $0.5\,\mathrm{ppmv}$ in the case of methane using our currently established, continuous $CH_4$ analysis and carbon dioxide, respectively). However, these biases are small compared to the dampening experienced by the record due to firn smoothing.

*Copyright statement.* TEXT

# 1   Introduction

The East Antarctic Plateau is characterized by low temperatures and low accumulation rates. This creates the conditions for the presence of very old ice near the domes in this region (Raymond, 1983; Martín and Gudmundsson, 2012). Thanks to this particularity, the oldest, undisturbed ice retrieved at Dome C within the EPICA project has been dated to $800,000$ years in the past (Bazin et al., 2013; Veres et al., 2013). The analysis of the ice, and of the bubbles within, has made the reconstruction of the Earth's past temperatures and atmospheric concentrations in major greenhouse gases over the last eight glacial cycles possible (Lüthi et al., 2008; Loulergue et al., 2008). In turn, this knowledge helps us to better understand the Earth climate system and its past and future evolutions (Shakun et al., 2012). Furthermore, there is currently an active search for ice as old as one-and-a-half million years within the European Beyond EPICA-Oldest Ice project. Most of the potential drilling sites for such old ice are located on the East Antarctic Plateau (Fischer et al., 2013; Van Liefferinge and Pattyn, 2013; Passalacqua et al., 2018). Within the next decade, we might therefore expect the retrieval of new deep ice cores at low accumulation drilling sites of East Antarctica, with ages reaching back to one-and-a-half million years or more.

However, the gas records in low accumulation ice cores cannot be interpreted as perfect records of the atmospheric history. Indeed, due to the process of gas trapping in the ice, two distinct effects create discrepancies between the actual atmosphere's history and its imprint in the ice. The first one is due to the heterogeneous structure of the firn when transforming into airtight ice with bubbles. The overall structure of the firn column is characterized by a progressive increase in density with depth, associated with the constriction of the interstitial pore network (Stauffer et al., 1985; Arnaud et al., 2000; Salamatin et al., 2009). At high enough densities, pores in the firn pinch off and isolate the interstitial air from the atmosphere (Stauffer et al., 1985). However, firn is a highly stratified medium (Freitag et al., 2004; Fujita et al., 2009; Hörhold et al., 2012; Gregory et al., 2014) and some especially dense strata (respectively less dense strata) might experience early (respectively late) pore closure when compared to the rest of the firn (Etheridge et al., 1992; Martinerie et al., 1992; Fourteau et al., 2019). Consequently, abnormal strata might enclose older (respectively younger) air than their immediate surroundings, creating age irregularities in the gas record (Mitchell et al., 2015; Rhodes et al., 2016). In turn, the irregularities appear as spurious values in the measured gas record (Rhodes et al., 2016; Fourteau et al., 2017). These anomalies have been referred to as layered gas trapping artifacts and do not reflect actual atmospheric variations. The second effect that creates differences between the atmosphere and its imprint in the ice is due to the combination of diffusive air mixing in the firn (Schwander, 1989) and the progressive closure of pores in a firn stratum (Schwander et al., 1993; Mitchell et al., 2015). Consequently, the air enclosed in a given ice layer does not originate from a single point in time, but is rather characterized by a continuous age distribution covering tens to

hundreds of years (Schwander et al., 1988; Schwander et al., 1993; Rommelaere et al., 1997). Therefore, the concentration measured in an ice stratum is an average of atmospheric concentrations over a period of time. This effect removes the fast variability from the record, and has therefore been referred to as the smoothing effect (Spahni et al., 2003; Joos and Spahni, 2008; Köhler et al., 2011; Ahn et al., 2014; Fourteau et al., 2017). The degree of alteration between the actual atmospheric history and the signal recorded in ice cores is strongly dependent on the local accumulation rate, with low accumulation ice cores being particularly affected both in terms of layered trapping artifacts (Rhodes et al., 2016; Fourteau et al., 2017) and in terms of smoothing (Spahni et al., 2003; Joos and Spahni, 2008; Köhler et al., 2011; Ahn et al., 2014; Fourteau et al., 2017). For layered gas trapping, Fourteau et al. (2017) report layering artifacts reaching up to $50\,\mathrm{ppbv}$ in the Vostok methane record during the Dansgaard-Oeschger event 17 period. For smoothing, Spahni et al. (2003) report a gas age distribution in the EPICA Dome C ice core dampening atmospheric variability faster than a few hundred years. Similar degrees of smoothing have also been reported by Köhler et al. (2015) and Fourteau et al. (2017) for the ice cores of Dome C and Vostok, respectively.

In order to properly evaluate the composition of past atmospheres based on the gas records in polar ice cores, it is thus necessary to characterize the specificities of those two effects. Gas age distributions can be calculated for the purpose of estimating firn smoothing. In the case of modern ice cores this may be accomplished by using gas trapping models parametrized by firn air and pore closure data (Buizert et al., 2012; Witrant et al., 2012). However, to estimate the gas age distribution in bubbles it is necessary to use a depth-profile of the progressive closure of pores in the firn, quantifying the transformation of open pores into closed bubbles. Due to the re-opening of closed pores at the surface of the firn samples used for porosity measurements, such profiles of pore closure are associated with large measurement uncertainties (Schaller et al., 2017; Fourteau et al., 2019). Moreover, a specific problem arises in the case of glacial ice cores from East Antarctica. They were formed under very low temperatures and accumulation rates, below $2\,\mathrm{cm\,ice\,equivalent\,yr^{-1}}$ ($\mathrm{cm\,ie\,yr^{-1}}$, Veres et al., 2013; Bazin et al., 2013), that are not encountered on the present-day Antarctic and Greenland ice sheets. Due to this modern analogue problem, it is not possible to sufficiently constrain gas trapping models, prohibiting robust estimation of the gas age distributions that were responsible for smoothing during glacial periods.

Layered gas trapping, on the other hand, originates from firn heterogeneities and is a stochastic process. Moreover, current gas trapping models do not fully represent the centimeter scale variability of the firn. The proper modelling of layered gas trapping is thus limited by the lack of knowledge of glacial firn heterogeneities and the difficulty to model gas trapping in a layered medium. Fourteau et al. (2017) proposed an empirical model to reproduce the artifacts observed in the Vostok ice core during the Dansgaard-Oeschger event 17. However, this model has only been applied to a single event and might not be directly

applicable for different periods and conditions.

For this work, we analyzed methane concentrations in five ice core sections from the East Antarctic Plateau. These sections cover accumulation and temperature conditions representative of East Antarctica for both the glacial and inter-glacial periods. The new data are used to test the layered gas trapping model proposed by Fourteau et al. (2017), with the goal of rendering the model applicable to a range of accumulation rates characteristic of glacial and inter-glacial conditions on the East Antarctic Plateau. Then, the gas age distributions for each of the five records are estimated by comparing the low accumulation records with much higher accumulation records (Fourteau et al., 2017). Indeed, gas trapping occurs faster at high-accumulation sites, which are thus less affected by both smoothing and layering artifacts and can therefore be used to produce atmospheric scenarios with low levels of alteration. Finally, we simulate gas trapping in a hypothetical million-and-a-half years old and thinned ice core, including layering artifacts. We then simulate the process of methane measurements using our continuous flow analysis system, and of carbon dioxide measurements using discrete measurements. The synthetic signals are finally compared to the original atmospheric reference to quantify the deterioration of atmospheric information due to firn smoothing and layered gas trapping.

## 2 Methods

### 2.1 Choice and description of the studied ice cores

The newly measured ice core sections originate from the East Antarctic sites of Vostok, Dome C, and Lock-In. The three sites are displayed on the map in Figure 1. The five measured sections correspond to time periods where high resolution methane measurements from high-accumulation ice cores are also available.

Modern Lock-In ice core:

The first studied ice core section is the upper part of the Lock-In ice core. The site of Lock-In is located $136\,\mathrm{km}$ away from Dome C, towards the coast ($3209\,\mathrm{m}$ above sea level, coordinates $74°08.310'$ S, $126°09.510'$ E). The local accumulation is $3.9\,\mathrm{cm\,ie\,yr^{-1}}$ (Yeung et al., 2019). About $80\,\mathrm{m}$ of ice was analyzed for methane, ranging from $116\,\mathrm{m}$ (near the firn-ice transition) to $200\,\mathrm{m}$ depth. Firn air sampling during the drilling operation was conducted down to $108.3\,\mathrm{m}$ depth. The gas record ranges from about $400$ to $3000\,\mathrm{yr\,BP}$ (Before Present, with present defined as 1950 CE in this article).

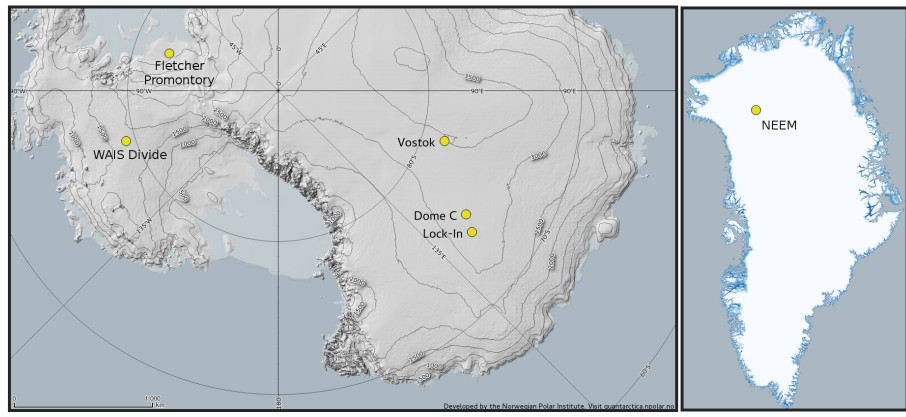

**Figure 1.** Left: Map of Antarctica with the sites of Lock-In, Dome C, Vostok, WAIS Divide, and Fletcher Promontory shown (made with the Quantartica package). Right: Map of Greenland with the site of NEEM shown (using Greenland Ice Mapping Project data; Howat et al., 2014).

Modern Dome C ice core:

A shallow ice core from Dome C was analyzed for depths ranging from $108$ to $178\,\mathrm{m}$. Similarly to Lock-In, this section corresponds to ages ranging from about $400$ to $3000\,\mathrm{yrBP}$, but is characterized by a lower accumulation rate of $2.7\,\mathrm{cm\,ie\,yr^{-1}}$ (Gautier et al., 2016).

Holocene Dome C ice core:

Ice from the second drilling of the EPICA Dome C ice core (referred to as EDC99 hereafter) was measured from $312$ to $338\,\mathrm{m}$ depth. The gas ages range from $7800$ to $8900\,\mathrm{yrBP}$, with an average accumulation of $3.1\,\mathrm{cm\,ie\,yr^{-1}}$ (Bazin et al., 2013; Veres et al., 2013). This period includes the $8.2\,\mathrm{ka}$ cold event (Thomas et al., 2007), notably characterized by a sharp decrease in

10   global methane concentrations (Spahni et al., 2003; Ahn et al., 2014).

DO6-9 Dome C ice core:

A section from the first drilling of the EPICA Dome C ice core (referred to as EDC96 hereafter) was analyzed for depths ranging from $690$ to $780\,\mathrm{m}$. The AICC2012 chronology indicates gas ages covering the period $33000$ to $41000\,\mathrm{yrBP}$, and an

15   average accumulation rate of $1.5\,\mathrm{cm\,ie\,yr^{-1}}$ (Bazin et al., 2013; Veres et al., 2013). The $CH_4$ excursion associated with the Dansgaard-Oeschger (DO) events 6 to 9 are included in this gas record (Huber et al., 2006; Chappellaz et al., 2013).

**Table 1.** Summary of the different ice core sections studied in the article with their associated atmospheric references.

| Ice Core Section | Coordinates | Accumulation $(\mathrm{cm\,ie\,yr}^{-1})$ | Atmospheric Reference |
|---|---|---|---|
| Lock-In Modern | 74°08′S 126°10′E | 3.9[a] | WAIS Divide discrete[b] |
| Dome C Modern | 75°06′S 123°20′E | 2.7[c] | WAIS Divide discrete[b] |
| Dome C 8.2ka | Same as above | $3.1 \pm 0.15$[d] | Fletcher-EDC composite[e] |
| Dome C DO6-9 | Same as above | $1.5 \pm 0.15$[d] | WAIS Divide CFA[f] |
| Vostok D021 | 78°28′S 106°50′E | $1.5 \pm 0.05$[d] | Deconvoluted NEEM CFA[g] |

a: Yeung et al. (2019); b: Mitchell et al. (2013); c: Gautier et al. (2016); d: Bazin et al. (2013); Veres et al. (2013), uncertainties given as standard deviation over the section; e: EDC data from Loulergue et al. (2008) used for the lower part of the composite; f: Rhodes et al. (2015); g: Chappellaz et al. (2013)

DO21 Vostok ice core:

The last analyzed section originates from the Vostok 4G ice core, for depths between $1249$ and $1290\,\mathrm{m}$. The expected gas ages span from $84000$ to $86500\,\mathrm{yrBP}$, with an average accumulation of $1.5\,\mathrm{cm\,ie\,yr}^{-1}$ (Bazin et al., 2013; Veres et al., 2013). This section was chosen as it includes the record of the $\mathrm{CH_4}$ excursion associated with the DO21 event, the fastest methane increase
of the last glacial period (Chappellaz et al., 2013).

**2.2   High-resolution methane measurements**

The five ice core sections were analyzed for methane concentrations using a Continuous Flow Analysis (CFA) system, including a laser spectrometer based on optical-feedback cavity enhanced absorption spectroscopy (OF-CEAS; Morville et al., 2005), at the Institut des Géosciences de l'Environnement (IGE), Grenoble, France. The laser spectrometer was calibrated to the
NOAA2004 scale (Dlugokencky et al., 2005) using three synthetic air standards of known concentrations. Spectra are acquired at a rate of $6\,\mathrm{Hz}$, and these spectra are later averaged to produce one concentration value per second. The five ice core sections were melted at an average rate of $3.6\,\mathrm{cm\,min}^{-1}$. For a similar set-up, Fourteau et al. (2017) showed that the CFA system is able to resolve variations down to the centimeter-scale ($50\,\%$ of attenuation for sine variations with a wavelength of $2.4\,\mathrm{cm}$). Yet, the measured concentrations are affected by the preferential dissolution of methane compared to nitrogen and oxygen in the
meltwater (Chappellaz et al., 2013; Rhodes et al., 2013). It is therefore necessary to apply a correction factor to account for the solubility effect. However, this factor is a priori not known and potentially differs between ice core measurement campaigns depending on factors such as the air content of the ice, or the precise CFA set-up. The methodology for correcting for methane dissolution is addressed in Section 3.1 below. Using a mixture of de-ionized water and standard gases, the analytical noise of the CFA system has been determined to be about $10\,\mathrm{ppbv}$ peak-to-peak (Fourteau et al., 2017).

The obtained records present numerous gaps, ranging from a few centimeters to several meters. Several reasons explain the presence of such gaps. First, the space between consecutive melting sticks let modern air enter the CFA system, resulting in a contamination and abnormally high methane concentrations. Moreover, the presence of cracks and fractures in the ice might also let modern air enter the measured ice stick itself, also resulting in abnormally high concentrations. The moments of potential air intrusions were recorded during the measurement campaigns, and the data were screened to remove the resulting contaminations, creating gaps in the record (Fourteau et al., 2017). Finally, some of the ice was simply not available for this study, resulting in further gaps in the records. This notably explains the gap visible around the $1260\,\mathrm{m}$ depth in the Vostok record (part C of Figure 4).

## 3 Results and Discussion

### 3.1 Correction for methane dissolution in the meltwater

In order to correct the measured mixing ratios for the preferential dissolution of methane in water, a correction factor is applied to the data to raise them to absolute values (Chappellaz et al., 2013; Rhodes et al., 2013). One way to estimate this correction factor is to compare the dissolution-affected CFA data with records that are already on an absolute scale.

The discrete methane measurements from the high accumulation WAIS Divide (WD) ice core published by Mitchell et al. (2013) cover the period from $500$ to $3000\,\mathrm{yr\,BP}$. We therefore use the WD data to estimate the correction factors for the modern sections of Lock-In and Dome C.

Loulergue et al. (2008) performed discrete methane measurements in the Dome C EPICA ice core, that include the $8.2\,\mathrm{ka}$, DO6 to 9, and DO21 events. We use these discrete methane data to correct our CFA data for the dissolution, for the Holocene and DO6-9 Dome C ice core sections and for the DO21 event section in Vostok. The comparison between the already calibrated methane dataset of Loulergue et al. (2008) and our corrected CFA measurements is given for the DO6 to 9 period in Figure 2. Note that in the case where our CFA data and the independent absolute measurements do not originate from the same ice core, the match between the datasets are performed on sections with low methane variability to reduce the influence of firn smoothing. The signals after correction for the dissolution of methane are displayed in light blue in the upper panels of each part of Figures 3 and 4. The gas age chronologies indicated in the figures are described in Section 3.4.1 of this article. The black superimposed curves in Figure 4 correspond to the part of the signal cleaned for layering artifacts, as described in Section 3.3.3. The correction factors for the solubility effect range between $1.12$ and $1.14$. They are close to the value of $1.125$ reported by Fourteau et al. (2017) for the Vostok ice core with the same IGE CFA system, but larger than the $1.079$ correction factor

reported by Rhodes et al. (2013) for the Greenland NEEM ice core with a different CFA system. This difference could to be due to the larger air content in NEEM as well as the different CFA systems used, including the melthead geometry or the specific setup of gas separation from the melt stream.

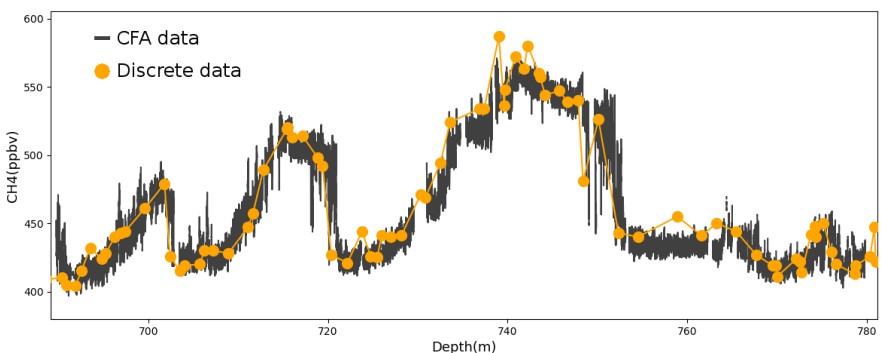

**Figure 2.** Illustration of a corrected CFA record (in black) by matching an already-calibrated record (orange dots, Loulergue et al., 2008).

## 3.2 Atmospheric references

In Sections 3.3.2 and 3.4.1 below, weakly smoothed methane records will be used as atmospheric references to study the layering artifacts and smoothing of low accumulation ice cores. To derive the atmospheric references we used methane gas records from higher-accumulation ice cores where higher frequency variations are much better preserved compared to the low-accumulation ice cores (Fourteau et al., 2017). When possible, we use high-accumulation records from ice cores drilled in Antarctica. Otherwise, in order to compare the high and low-accumulation records we need to estimate the methane inter-
hemispheric gradient.

For the modern Lock-In and Dome C sections, we used the discrete measurements from the WD ice core, also used for correcting the methane preferential dissolution (Mitchell et al., 2013). For the Holocene section of EDC99, we used CFA data from an ice core drilled at Fletcher Promontory as the atmospheric reference (Robert Mulvaney and Xavier Faïn, personal communication), complemented by discrete EDC data from Loulergue et al. (2008) for the oldest part of the period (depths below
330 m in part A of Figure 4). Using the low-accumulation EDC99 record for the lowest part does not deteriorate the quality of the composite for our application, as this part does not include fast atmospheric variations, and using a low-accumulation record is therefore appropriate in this specific case. The dating of the Fletcher-EDC99 composite was made consistent with the WAIS Divide chronology by manually selecting tie points (Buizert et al., 2015). For the DO6-9 events section in the EDC96 ice core, we used high-resolution CFA data from the WD ice core (Rhodes et al., 2015). The WD gas record dating has been

made consistent with the AICC2012 and GICC chronologies following Buizert et al. (2015). Finally, the atmospheric reference used for the Vostok DO21 period is based on the Greenland NEEM CFA data published by Chappellaz et al. (2013). Chappellaz et al. (2013) propose two CFA records obtained with two different spectrometers. For this study, we used the average values of the two instruments. As the Vostok and NEEM sites are located in different hemispheres, it is necessary to take into account the inter-hemispheric methane gradient between the two sites to make the methane variability of the two records as consistent as possible. Using the Vostok and NEEM records, we evaluated this inter-hemispheric gradient to be $30\,\mathrm{ppbv}$ for the DO21 period. This value is in line with the work of Dällenbach et al. (2000), and has been corroborated using the EDML methane record (EPICA Community Members et al., 2006). Yet, it remains possible that the inter-hemispheric gradient was not constant during this period of global methane rise. The use of a high resolution and weakly smoothed Antarctic record as the atmospheric reference would resolve this inter-hemispheric gradient problem, but such a record is not available. The original data of Chappellaz et al. (2013) present various gaps, that were filled by computing a splined version of the original NEEM CFA record. The spline was chosen not to induce smoothing in the NEEM record and to overlap with the original dataset in parts where data already existed. It is shown together with the original Chappellaz et al. (2013) data in Figure S1 of the Supplement.

Yet, we observed that the NEEM record cannot be directly used as an atmospheric input for the DO21 period. As explained in Section S1 of the Supplement, the first feature of the DO21 event (around the $1260\,\mathrm{m}$ depth in part C of Figure 4) is partially smoothed in the NEEM record. To retrieve the full amplitude of this fast event, we used the deconvolution technique described in Witrant and Martinerie (2013) and Yeung et al. (2019). To be applicable this method needs as input the gas age distribution responsible for the smoothing of the NEEM record. For this, we chose the age distribution estimated with a gas trapping model at the modern site of Siple Dome which has an accumulation $10.8\,\mathrm{cm\,ie\,yr^{-1}}$ (Witrant et al., 2012), similar to the accumulation of NEEM during the DO21 period (around $11.3\,\mathrm{cm\,ie\,yr^{-1}}$ Rasmussen et al., 2013). In order to test the sensitivity of our results to the deconvolution step, we performed several deconvolutions with age distributions of modern sites with accumulation values above and below the range of potential NEEM accumulations during the DO21 period. The results are presented in Section S1 of the Supplement and indicate that the deconvolution of NEEM is well constrained, and only weakly depends on the specific choice of the age distribution. The effect of the deconvolution is mainly to increase the amplitude of the fast feature at the onset of the DO21, which in turn increases the consistency between the NEEM and Vostok methane records. The drilling sites of the atmospheric references are displayed in Figure 1.

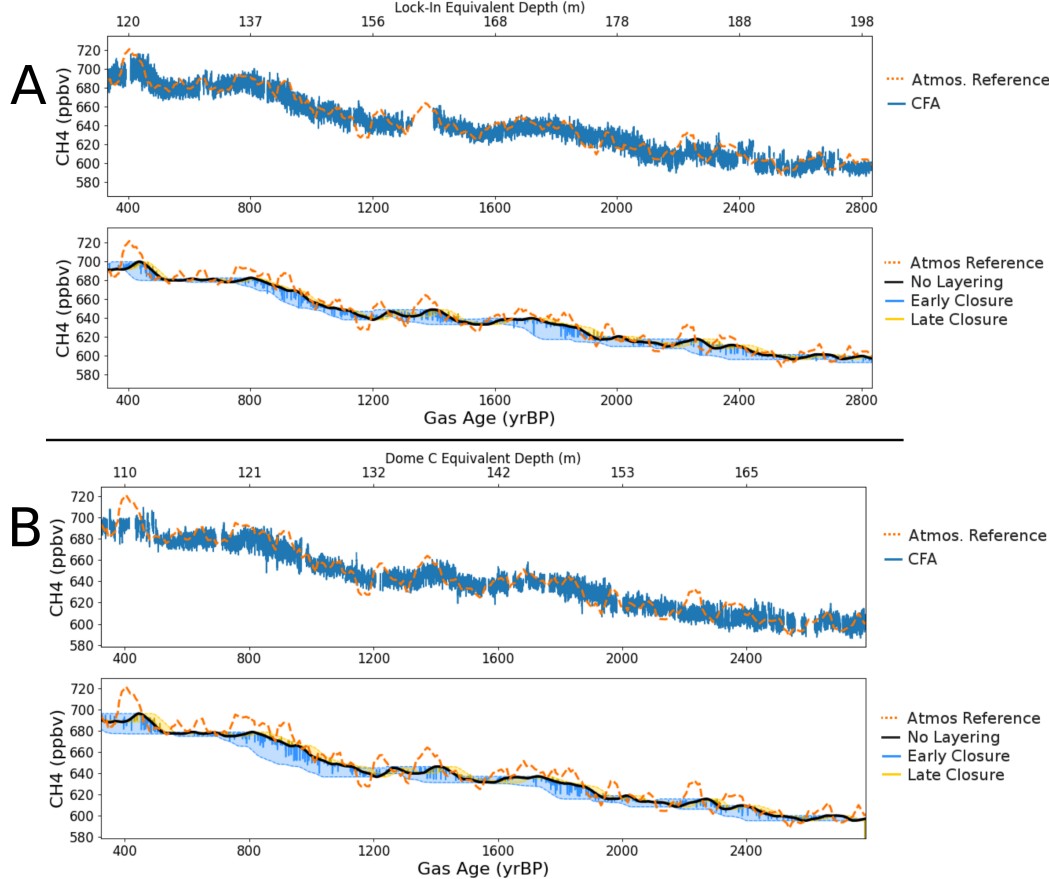

**Figure 3.** Measured and modeled modern Lock-In (part A) and Dome C (part B) methane profile. For each ice core, the upper panel shows the high resolution methane measurements in light blue, and the atmospheric reference in orange. The lower panel shows the layering artifacts model results. The black curve is the expected signal without artifacts. The blue and yellow spikes are the randomly distributed modeled early and late closure artifacts, respectively. The blue and yellow shaded areas respectively correspond to the modeled range of early and late artifacts with density anomalies up to two standard deviations. The dashed orange curve is the record used as the atmospheric reference.

## 3.3 Layered trapping artifacts

It has been observed in the high-accumulation firn of DE08 that not all layers close at the same depth (Etheridge et al., 1992), and that the air in summer layers is generally older than in winter layers. Building on this idea, Mitchell et al. (2015) and Rhodes et al. (2016) proposed that the layering observed at the firn-ice transition induces gas stratigraphic irregularities and age inversions in the record. Rhodes et al. (2016) and Fourteau et al. (2017) used such stratigraphic heterogeneities to explain the observation of sudden variations in high resolution methane records. These abrupt methane variations do not correspond to actual atmospheric variations and have been called layered trapping artifacts, or more simply layering artifacts. The mechanism

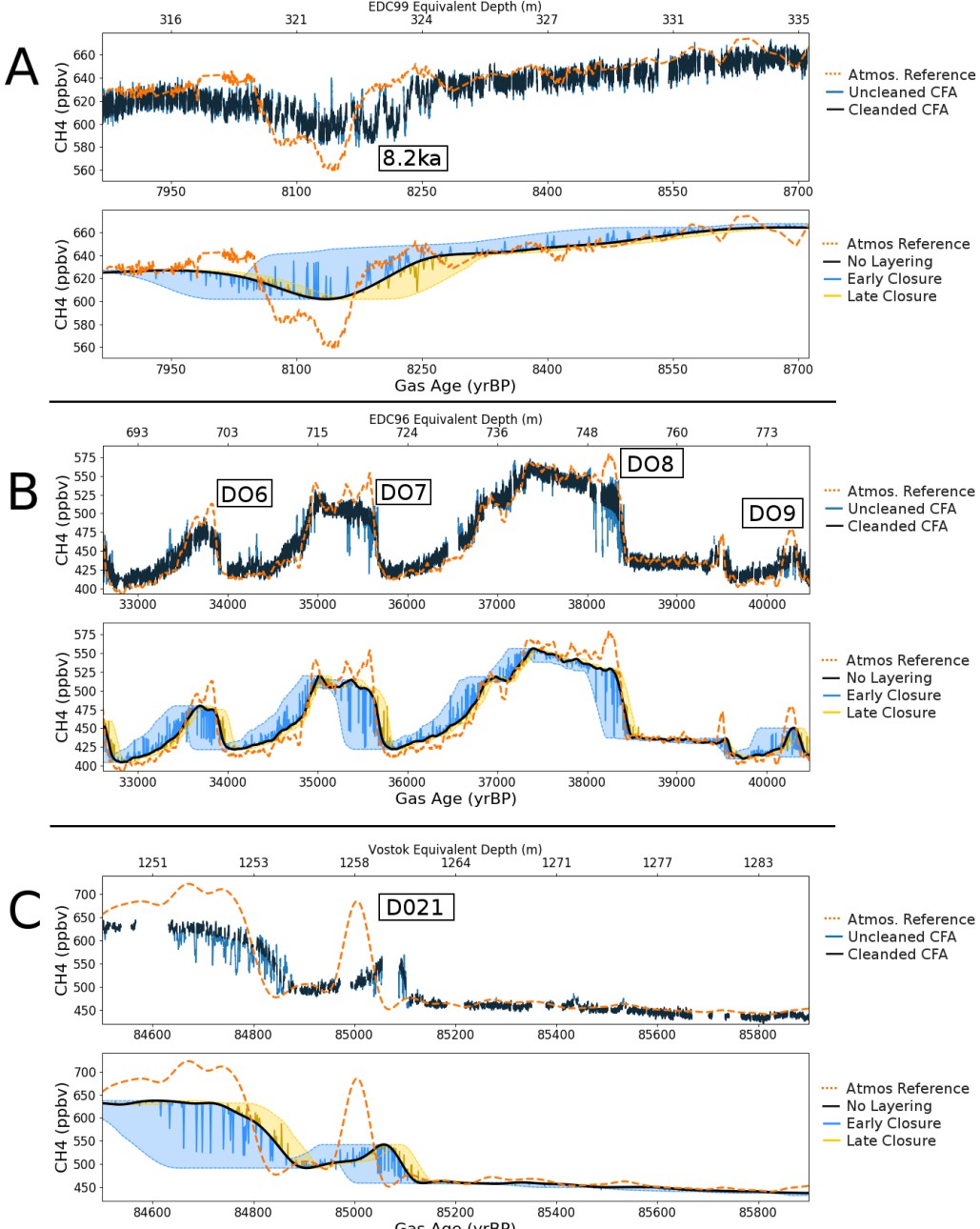

**Figure 4.** Same as Figure 3 for the Holocene EPICA Dome C (part A), EPICA Dome C DO6-9 (part B), and Vostok DO21 (part C) sections. In addition the superimposed black signals in the upper panels show the data cleaned of layering artifacts.

producing them is that layers closing especially early (or late) contain air older (or younger) than their immediate surroundings. During periods of variations of atmospheric composition, the methane mixing ratios in these layers are thus different than in

their surroundings, and appear as anomalous values (Rhodes et al., 2016). This also implies that $CH_4$ trapping artifacts are only observed in periods during which atmospheric $CH_4$ concentrations change on centennial time scales.

In accordance with Rhodes et al. (2016) and Fourteau et al. (2017), we observe abrupt centimeter-scale variations in the records during periods when the atmospheric $CH_4$ concentration was varying, that we interpret as layering artifacts. We visually identified such layering artifacts as spikes with widths of a few centimeters and whose concentrations are larger than the analytical noise of $\sim 10\,\mathrm{ppbv}$. Despite the effort to clean the record of modern air contaminations, it is possible that some contamination spikes remain and would then be wrongly interpreted as layering artifacts. However, the presence of negatively oriented spikes that cannot be attributed to contamination, and the occurrence of the spikes near periods of fast atmospheric methane variations confirm that these centimeter-scale abrupt variations are mostly due to the mechanism of layered gas trapping. They are notably well marked during the onset of DO events 8 and 21, corresponding to $750$ and $1252\,\mathrm{m}$ depth in the EDC96 and Vostok ice cores, respectively, in parts B and C of Figure 4. A zoom of the EDC96 record is available in Figure S3 of the Supplement to display examples of individual layering artifacts. They appear as reduced methane concentration layers near the onset of the DO events. The layered trapping artifacts tend to disappear in the absence of marked atmospheric variations. This explains the absence of important layering artifacts in the Lock-In and Dome C modern records (both parts of Figures 3).

The usage of high-resolution measurements allow us to easily distinguish layering artifacts as abrupt spikes exceeding the analytical noise in the record. However, with a lower-resolution technique, such as discrete measurements, it is possible to inadvertently measure a layering artifact without realizing that it is not representative of its surrounding concentrations, which would result in an anomalous point in the record. As a specific example, Fourteau et al. (2017) pointed out that one of the data points of the DO8 event in the EPICA Dome C methane record published by Loulergue et al. (2008) might have been sampled in an early closure layer, and thus represents a spurious value in their record. By comparing our new high-resolution record with the one of Loulergue et al. (2008) in Figure 5, we confirm that this data point does not correspond to atmospheric variability, and that it was sampled in a zone with a high number of early closure artifacts. In the case where such an abrupt variation is observed with discrete measurements, an additional sample should be measured in the vicinity of the first sample in order to confirm that the variability is not due to layered gas trapping.

Important late closure artifacts are unusual in methane gas records. The new measurements confirm this observation. Late closure artifacts, that should appear as positive methane anomalies at the onset of the DO events, are almost absent in our measurements. This can be explained by the fact that the surrounding firn layers are already sealed and prevent gas transport to greater depths, which in turn prevents the late closure layers from enclosing young air (Fourteau et al., 2017).

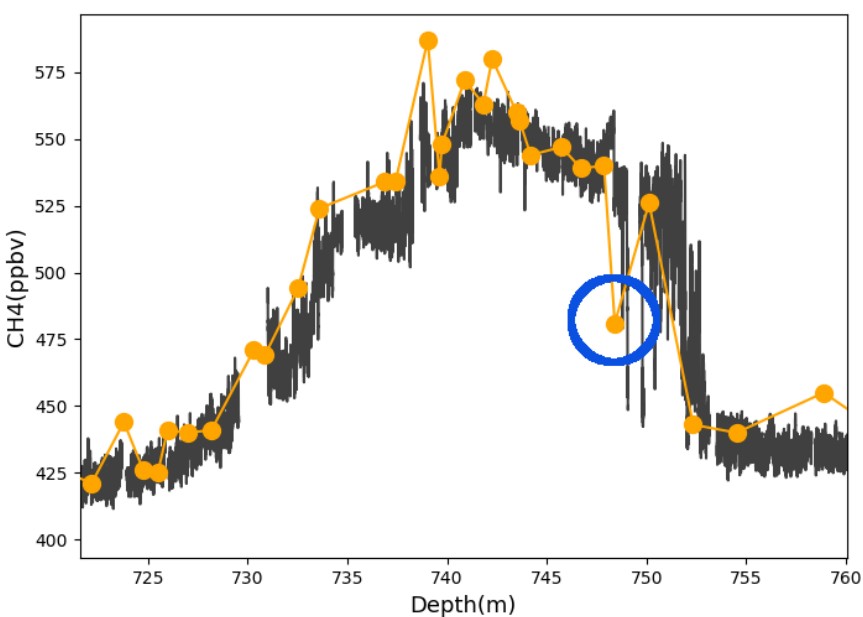

**Figure 5.** In orange: discrete methane measurements of the DO8 event in the EPICA Dome C record (Loulergue et al., 2008). In black: continuous methane measurements of the same event in the EPICA Dome C record (this study). In blue we highlight a data point of the discrete record likely corresponding to an early closure artifact.

### 3.3.1 Impact of chemistry on layered gas trapping

The EDC96 methane data suggest a lack of layering artifacts affecting the DO6 event (around $700\,\mathrm{m}$ depth in part B of Figure 4). We counted only 14 layering artifacts that clearly stand out of the analytical noise on the $20\,\mathrm{m}$ long section between 695 and $715\,\mathrm{m}$. For a comparison, we counted 15 artifacts on the $4\,\mathrm{m}$ long section from 718 to $722\,\mathrm{m}$ (DO7 event). These sections are not directly comparable as they did not undergo the same atmospheric methane variations during bubble enclosure, but it nonetheless suggests that the upper section of the DO6-9 record shows a lower number of layering artifacts per meter. Previous studies have highlighted the role of chemical impurities for the presence of dense strata in polar firn. Hörhold et al. (2012) and Fujita et al. (2016) argue that the presence of ions can soften the firn and facilitate the densification of some strata.

To evaluate the potential impact of ions on layered gas trapping artifacts in the EDC96 methane record, we compared the high resolution methane data with total calcium data measured in the ice phase (Lambert et al., 2012). We chose to focus on calcium since high resolution data are readily available and since Hörhold et al. (2012) observed a correlation between calcium and density variability in deep firn. However, as pointed out by Hörhold et al. (2012) and Fujita et al. (2016), this does not imply that calcium is the ion responsible for the establishment of deep firn stratification. Indeed, calcium is correlated with other ion

species that could be the cause for the preferential densification of some firn strata (Fujita et al 2016). A comparison of the calcium and methane datasets is shown in Figure 6. The depth difference between the EDC96 ice core, in which the methane measurements were performed, and the EDC99 ice core, in which the calcium measurements were performed, was taken into account using 18 individual volcanic markers (Parrenin et al., 2012). The calcium data clearly show a transition between two parts: ice below the depth of 718 m is characterized by relatively high calcium values and a high variability, while ice above 718 m is characterized by low calcium values and a low variability. The transition between these two parts corresponds to the high methane plateau of the DO7 event. It also matches with the transition described above between the two regions of large and small numbers of layering artifacts. It therefore appears that the number of significant artifacts (standing out of the analytical noise) may be correlated to the variability of calcium concentrations. While this preliminary result appears to support the idea of ion-induced layering artifacts, further work is still required on this question.

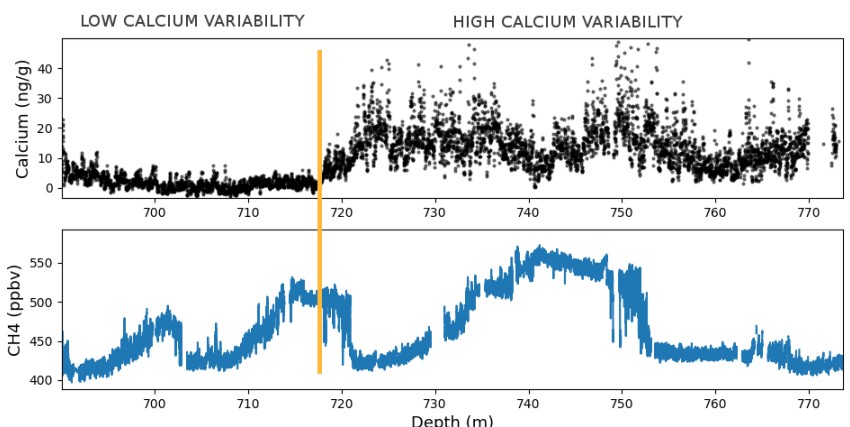

**Figure 6.** Upper panel: calcium concentrations measured in the EPICA Dome C ice core section covering the DOs 6 to 9 (Lambert et al., 2012). Lower panel: methane mixing ratios measured in this study.

### 3.3.2 Modeling layered trapping artifacts

The ability to model layered gas trapping helps quantify and predict its impact on gas records. For this purpose, we revised the simple layered gas trapping model proposed by Fourteau et al. (2017) for the Vostok DO17 section. Our aim is to parametrize the model in order to reproduce the layering artifacts observed in the new five measured sections, as well as in the previously published DO17 section of the Vostok core. This layered trapping model is based on the physical assumption that the density anomaly of a layer (its density difference relative to the meter-scale density profile, once centimeter-scale variability is

removed) can directly be converted into a trapping depth anomaly (the difference in closing depth compared to the average behavior). The depth anomalies are given by:

$$\Delta z = \frac{\Delta \rho}{\partial_z \rho} \qquad (1)$$

where $\Delta z$ is the depth anomaly of a given layer, $\Delta \rho$ its density anomaly, and $\partial_z \rho$ the derivative of the meter-scale firn density with depth in the trapping zone. For the rest of the article $\partial_z \rho$ will be referred to as the "densification rate". The methane concentration in an early trapping layer is then simply the concentration that would normally be trapped at a depth $\Delta z$ below. The original Fourteau et al. (2017) model was based on the computation of the age anomalies of abnormal layers (the time difference between the closure of the layers and the average behavior). However, this age-based formulation is sensitive to the dating of the ice core, and can lead to abnormally strong artifacts if the chronology is poorly constrained. On the other hand, the depth-based formulation proposed here (Equation 1) is not affected by dating uncertainties.

The model requires a densification rate for the firn-ice transition zone. Observations of density profiles in Dome C and Vostok proposed by Bréant et al. (2017) reveal densification rates around $2.2 \, \mathrm{kg \, m^{-3} \, m^{-1}}$ for both sites. This value also applies to the Lock-In firn as inferred from high resolution density measurements (Fourteau et al., 2019). Therefore, the densification rate at the firn-ice transition was set at $2.2 \, \mathrm{kg \, m^{-3} \, m^{-1}}$ for all sites studied here. Finally, the model requires a typical density anomaly in the closing part of the firn. Based on the linear relationships of density standard deviation with temperature and accumulation proposed by Hörhold et al. (2011), Fourteau et al. (2017) estimated the typical density anomaly to be $5 \pm 2 \, \mathrm{kg \, m^{-3}}$ for the Vostok firn during the glacial DO17 event. Moreover, Hörhold et al. (2011) report a density standard deviation of $4.6 \, \mathrm{kg \, m^{-3}}$ around the firn-ice transition of the Dome C site. As a first guess, we set the density anomalies $\Delta \rho$ to obey a zero-centered Gaussian distribution of standard deviation $5 \, \mathrm{kg \, m^{-3}}$. Similarly to Fourteau et al. (2017), the depth anomalies of late closure layers are reduced by $75\%$ in order to replicate their low impact in the measured records.

The depth ranges of expected modeled artifacts are displayed below the newly measured signals in Figures 3 and 4 (blue and yellow shaded areas), as well as the modeled artifacts for the DO17 event in Vostok in Figure 7. The model produces layering artifacts in the expected parts of the records (mostly right after the onsets of DO events) and with the expected sign and amplitude. This is visible on Figures 3, 4, and 7 where the layering artifacts of the record fall within the expected envelopes of the model results. However, it is hard to reproduce the precise distribution of layering artifacts. This is due to the fact that the position of layering artifacts is random, and that the observed layering artifacts in an ice core record are one single outcome

among many possibilities. Moreover our assumption of a Gaussian distribution of the density anomalies may also influence our results. Nonetheless, the simple model presented here provides reasonable results for accumulation conditions ranging from 3.9 to $1.3\,\mathrm{cm\,ie\,yr^{-1}}$, encompassing most of the climatic conditions of the East Antarctic Plateau for both the glacial and interglacial periods.

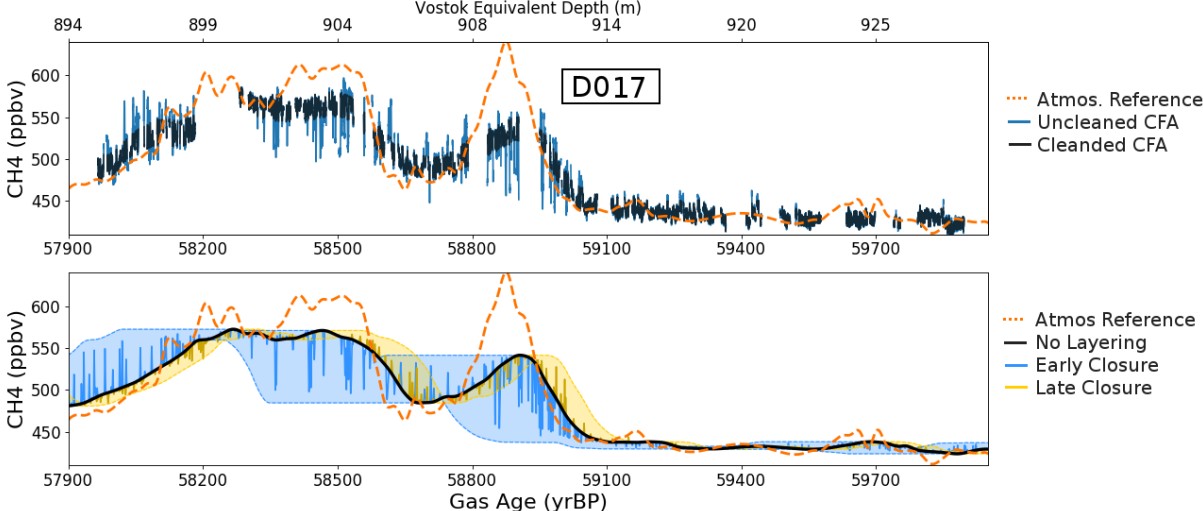

**Figure 7.** Same as Figure 4 for the DO17 Vostok section. The methane data of the upper panel are from Fourteau et al. (2017) and the atmospheric reference is from Rhodes et al. (2015).

### 3.3.3   Removing layered trapping artifacts

Since layered gas trapping artifacts are not due to chronologically ordered atmospheric variations, a proper interpretation of gas records requires them to be removed from the datasets. As pointed out by Fourteau et al. (2017), in some sections of the records these artifacts only exhibit negative or positive methane anomalies. They cannot therefore be removed by applying a running average without introducing bias in the signal.

As the layering artifacts can be visually distinguished, it would be possible to remove them manually from the data. However, such procedure would be cumbersome due to the large amount of CFA data points. Hence, to clean the new three methane signals presenting layering artifacts (EDC96, EDC99 and Vostok) we use a recursive cleaning procedure similar to the algorithm described by Fourteau et al. (2017). The cleaning algorithm starts by estimating a smooth signal that should represent the measured signal free of layering artifacts and analytical noise. For this purpose, a running median is first computed to remove the layering artifacts while minimizing bias. Then, the signal is smoothed with a binned average, and interpolated back

to high resolution using an interpolating spline. Then, the analytical noise is estimated using the Normalized Median Absolute Deviation of the data (NMAD, Rousseeuw and Hubert, 2011; Fourteau et al., 2017). The data are clipped above 2.5 times the value of the NMAD, in order to trim a part of the artifacts. The algorithm is repeated until the signal is determined to be free of layering artifacts. The signal is considered to be free of artifacts when the NMAD (estimation of noise without the layering artifacts) and the standard deviation (estimation of noise with the layering artifacts) are similar (Fourteau et al., 2017). This recursive method produces a progressive removal of the layering artifacts.

This procedure was successfully applied to the Holocene section of EDC99 that only exhibits a few clear artifacts near $323\,\mathrm{m}$ depth. The resulting cleaned signal is displayed in the upper panel of part A of Figure 4 as a black curve. Note that the cleaning algorithm also removes a small part of analytical noise. However, this small removal of the analytical noise is negligible and does not influence our conclusions.

Direct application of this algorithm to the EDC96 DO6-9, and Vostok DO21 records leads to an ineffective removal of layering artifacts in periods of fast methane rise. After detailed investigation, it appears that the main issue the algorithm is facing is the determination of an artifacts free signal by the running median. In some sections of the ice cores, the signal displays high numbers of artifacts with methane anomalies of $50\,\mathrm{ppbv}$ or more, as seen for instance in the onset of the DO8 in part B of Figure 4. Under these conditions, the running median is influenced by the presence of the artifacts and yields a biased signal. To circumvent this problem, we manually provided a set of values that correspond to the artifact free part of the signal. As it is easy to visually distinguish the layering artifacts from the rest of the signal, the manual selection of these points is fairly unambiguous and straightforward. The rest of the algorithm then proceeds as normal, which allows us to use the automated determination of analytical noise and cleaning of the data. This also ensures a better consistency of cleaning between the different CFA datasets. The cleaned EDC99 Holocene, EDC96 DO6-9 and Vostok DO21 signals are displayed in the upper panels of each part of Figure 4 in black.

## 3.4 Smoothing in East Antarctic ice cores

The five ice core sections measured in this article all originate from the East Antarctic Plateau. The low temperatures and aridity of this region result in a slow densification of the firn, and a slow bubble closure. Therefore, the gas enclosed in a given ice layer tends to have a broad age distribution. This leads to an significant smoothing of atmospheric fast variability observed in East Antarctic ice cores (Spahni et al., 2003; Joos and Spahni, 2008; Köhler et al., 2011; Ahn et al., 2014; Fourteau et al., 2017).

### 3.4.1 Estimating gas age distributions

Since smoothing in an ice core record is a direct consequence of the gas age distribution (GAD), the knowledge of GADs for various temperature and accumulation conditions is necessary to predict the impact of smoothing on gas signals. We thus apply the GAD extraction method proposed by Fourteau et al. (2017) to the five new high resolution records.

This method, designed to estimate the GAD of low accumulation records, is based on the comparison with a weakly smoothed record derived from a high accumulation ice core used as an input atmospheric reference (see Section 3.2). The idea of the method is to find a GAD that is able to smooth the atmospheric reference to the level of the low-accumulation record. We thus searched for the GAD that minimizes the Root-Mean-Square Deviation between the CFA measurements and the smoothed version of the atmospheric reference. In order to have a well-defined problem in a mathematical sense, the GAD of the low-accumulation ice core is assumed to be a log-normal function. Such a log-normal distribution is fully defined by two independent parameters. Finding the best GAD to smooth the atmospheric reference is then reduced to the recovery of a pair of optimal parameters. Nonetheless, log-normal distributions exhibit a large range of shapes that can adequately represent age distributions (Köhler et al., 2011; Fourteau et al., 2017). In order for the GAD extraction to perform well, it is also necessary to have a well defined gas age chronology for the low-accumulation record, so that its methane variability is well aligned with the atmospheric reference. This chronology is built recursively during the GAD extraction procedure by manually selecting tie-points, in order to be consistent with the already-existing chronology of the atmospheric reference (Fourteau et al., 2017). The GAD extraction procedure is designed to match the CFA record and the convoluted atmospheric reference only where CFA data are available. We thus do not have to extrapolate the CFA data within the gaps of the record. However, smaller gaps in the records would have reduced the uncertainty of the estimated GADs, as the additional data would have provided more constraints on the GAD estimation.

The gas age distributions obtained for the five new ice core sections are displayed as solid lines in Figure 8, and an uncertainty analysis is available in Section S3 of the Supplement. The parameters defining the different log-normal distributions are listed in Table 2. The standard deviation in this table should not be directly interpreted as the broadness of the age distribution and therefore its degree of smoothing. Indeed, some distributions exhibit strong asymmetries that tend to increase the standard deviation values. For instance, the Vostok DO21 age distribution has a standard deviation value much larger than the other distributions, but does not induce a much larger smoothing.

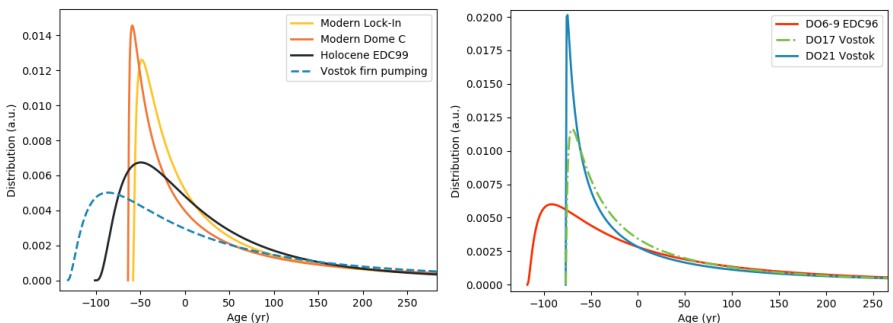

**Figure 8.** Gas age distributions in various ice cores. In order to be more easily compared, the age distributions are all drawn relative to their median gas age. This is why parts of the distributions have negative ages in the graph. The left panel displays Holocene GADs and the right panel glacial period GADs. The DO21 Vostok, DO6-9 EDC96, Holocene EDC99 and modern Lock-In and Dome C age distributions shown as solid lines were determined in this study. The dashed Vostok DO17 distribution was also determined by comparison with a high-accumulation record (Fourteau et al., 2017). The dashed Vostok firn pumping distribution was obtained with a gas trapping model, constrained with firn air pumping data (Witrant et al., 2012).

**Table 2.** Parameters defining the log-normal gas age distributions derived by comparison with weakly smoothed records. The location and scale parameters correspond to the $\mu$ and $\sigma$ parameters used to define the log-normal law: $\mathrm{GAD}(a) = \frac{1}{a\sigma\sqrt{2\pi}}\exp(-\frac{(\ln(a)-\mu)^2}{2\sigma^2})$, where $a$ is the age.

| Site and Period | Location | Scale | Mean (yr) | Std Dev (yr) |
|---|---|---|---|---|
| Lock-In Modern | 4.063 | 1.333 | 141 | 313 |
| Dome C Modern | 4.156 | 1.611 | 234 | 823 |
| Dome C 8.2ka | 4.618 | 1.100 | 185 | 285 |
| Dome C DO6-9 | 4.765 | 1.231 | 250 | 472 |
| Vostok DO21 | 4.339 | 1.988 | 552 | 3950 |

### 3.4.2 Smoothing and accumulation

The degree of smoothing in a gas record is strongly linked to the accumulation rate under which the gases are trapped, with low accumulation sites exhibiting a stronger degree of smoothing (Spahni et al., 2003; Joos and Spahni, 2008; Köhler et al., 2011). First, a lower accumulation is generally associated with a lower temperature. Low-temperature ice deforms less easily, slowing the densification process such that bubble closure spans a larger time period. Moreover, low accumulation is also associated with a slow mechanical load increase of the ice material, also resulting in a slower densification and a longer bubble closure period.

Our results suggest that, during the glacial period, East Antarctica ice cores are affected by the same level of smoothing, that can be represented with the same gas age distribution. We produced two new glacial period gas age distributions (EDC96 DO6-9 and Vostok DO21), that are to be added to the previously published GAD obtained for the Vostok site during the DO17 event using the same GAD extraction method ($1.3\,\mathrm{cm\,ie\,yr^{-1}}$ accumulation rate; Fourteau et al., 2017). These three age dis-

5 tributions of East Antarctic sites under glacial conditions are displayed in the right panel of Figure 8. The smoothing they induce is presented for the DO21 and DO6 to 9 events in parts A and B of Figure 9. It appears that the two Vostok distributions lead to a similar smoothing. On the other hand the GAD obtained for EDC96 for the DO6-9 events is significantly broader than the Vostok ones and results in a stronger smoothing, especially visible at the onset of the DO21 event. This is surprising as the Vostok DO21 and Dome C DO6-9 records have the same accumulation rate, and should therefore present similar age

distributions.

However, the uncertainty analysis in Section S3 of the Supplement reveals that the age distribution of the EDC96 record is poorly constrained. This indicates that the smoothing of the DO6-9 period is weakly sensitive to the choice of GAD, and therefore that a large range of GADs results in adequate smoothing for the EDC96 record. On the other hand, the DO21 period is very sensitive to the choice of GAD, and much fewer age distributions are able to reproduce the smoothing of the Vostok

DO21 record. Our understanding is that because of its shape and its fast atmospheric rate of change (Chappellaz et al., 2013), the first feature of DO21 is very sensitive to the choice of GAD, despite a gap in the record, and is therefore a good discriminant between age distributions. On the other hand, the step-like features of the DO6-9 record are less sensitive to the choice of GAD, which leads to a less-well constrained extraction of age distributions. Consequently, the distributions obtained for glacial Vostok (DO17 and 21 events) are also suited for the smoothing of DO6 to 9 events in EDC96, as seen in part B of Figure 9.

This suggests that the smoothing of the three glacial records are similar, and that the three ice cores therefore can be characterized by similar gas age distributions. We thus propose to use a common gas age distribution to represent the smoothing in ice cores with accumulations below $2\,\mathrm{cm\,ie\,yr^{-1}}$. The distribution proposed by Fourteau et al. (2017) is a good candidate for this common distribution, as it reproduces well the smoothing in the EDC96 DO6-9 record and the Vostok DO17 and 21 records. Moreover, its shape is a compromise between the two other glacial GADs proposed in this article (right panel of Figure8).

Our results also indicate that the glacial and inter-glacial smoothings of East Antarctic ice cores are relatively similar. Indeed, comparing the typical GAD of the glacial period with the ones obtained for the Holocene and modern periods indicates that the latter lead to a slightly smaller degree of smoothing. This is illustrated with the smoothing of the $8.2\,\mathrm{ka}$ event displayed in part C of Figure 9. It is consistent with the expectation that the smoothing during the low-accumulation glacial period

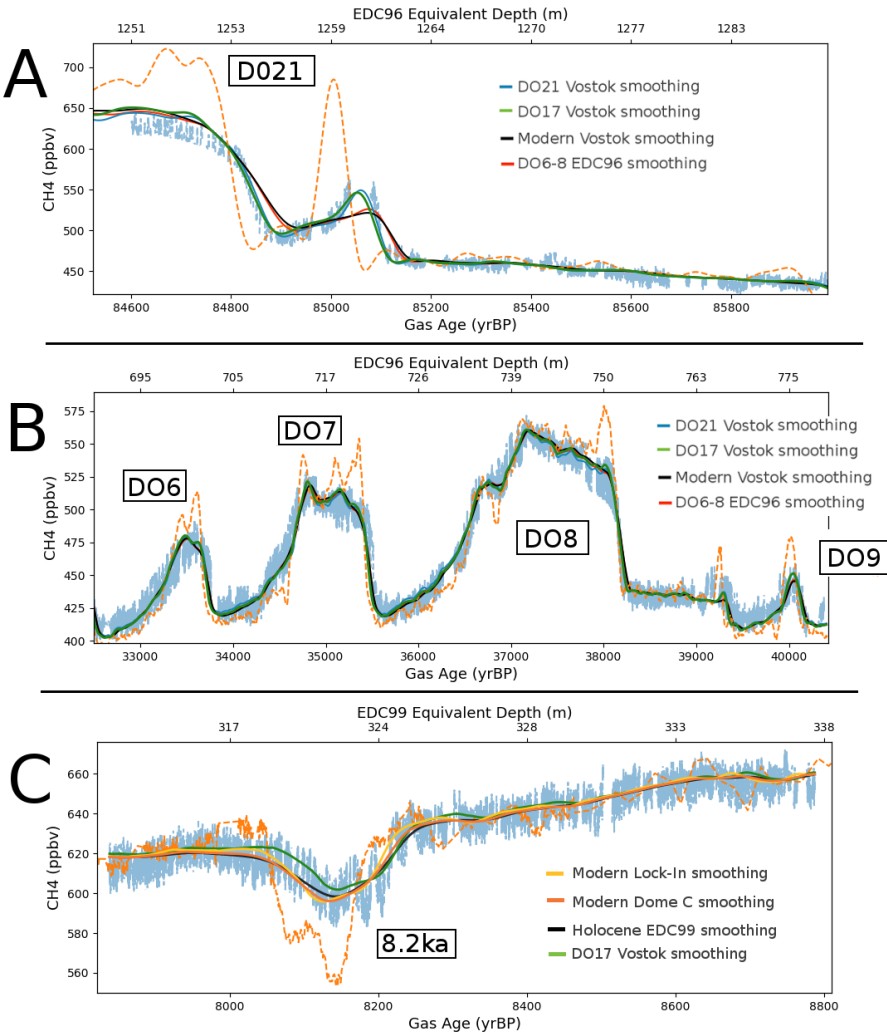

**Figure 9.** Parts A and B: comparison of smoothing resulting from for the three different glacial age distributions, and the modern Vostok GAD constrained with firn air pumping data. Atmospheric references (dashed lines) have been smoothed by the four different GADs and are displayed overlying the measurements (light blue). Part A shows the DO21 event as measured in Vostok, and part B shows the DO6 to 9 as measured in EDC96.

Part C: Comparison of the smoothing induced by three different Holocene and a glacial age distributions. The atmospheric reference for the 8.2 ka event (dashed line) has been smoothed by the four different GADs and displayed over the EDC99 measurements (light blue).

should be greater than during the higher-accumulation Holocene period. However, the doubling of accumulation between the DO17 event at Vostok and the 8.2 ka event in EDC99 does not result in a drastic decrease of the smoothing. It is also note-worthy that for the Dome C site, the transition from 8.2 ka conditions ($3.1\,\mathrm{cm\,ie\,yr^{-1}}$ accumulation rate) to modern conditions

$(2.7 \, \mathrm{cm \, ie \, yr^{-1}}$ accumulation rate) does not significantly change the degree of smoothing. It thus appears that for accumulations below $3.9 \, \mathrm{cm \, ie \, yr^{-1}}$ the degree of smoothing only slowly increases with the decrease of the accumulation rate.

Finally, the expected smoothing using the firn pumping GAD estimation for modern Vostok is also displayed in parts A and B of Figure 9 in black (Witrant et al., 2012). This GAD was obtained with a method independent of the GAD extraction technique of this article. The smoothing estimated for modern Vostok is stronger than the one measured in the glacial Vostok ice core. This can be seen at the onset of the DO21, in part A of Figure 9, where the smoothing induced by the Vostok firn pumping distribution results in a signal with a smaller magnitude than the CFA measurements. This confirms the observation of Fourteau et al. (2017) that the firn pumping GAD of modern Vostok leads to a stronger smoothing than the one observed during the DO17 event. This stronger smoothing in a modern ice core is in contradiction with the Holocene and modern GADs reported in this article for Dome C and Lock-In. This discrepancy could result from the large uncertainties associated with the firn-model-based GAD, which notably arise from the poor constraints on the closed porosity profiles used (Schaller et al., 2017). Unfortunately, high-resolution methane measurements of the late Holocene period in the Vostok ice core are not available to apply the GAD extraction method.

### 3.4.3 Effect of smoothing on record synchronization

Compared to the GADs of the Holocene and modern periods, the GADs of the glacial period exhibit a stronger degree of skewness (see Figure 8). This asymmetry of the log-normal distributions, with a long tails for old ages, means that the smoothing not only removes variability, but also induces phase shifts during fast variations (Fourteau et al., 2017). This is especially visible for the variation recorded at the depth of $1260 \, \mathrm{m}$ in the Vostok DO21 ice core section, displayed in part A of Figure 9. The peak of the event is recorded a couple of meters before its position in the absence of smoothing. In terms of ages, this corresponds to an error of about $85 \, \mathrm{yr}$ in the Vostok gas chronology. Studies relying on the synchronization of atmospheric variability between ice cores should be aware of this potential bias (Bazin et al., 2013; Veres et al., 2013). To produce consistent chronologies, the synchronization should be performed on signals with a similar degree of smoothing (either by convolving the higher-accumulation record or deconvolving the lower-accumulation one). Otherwise, the phase shifts should be taken into account in the age uncertainty estimates.

## 4 Loss of climatic information in a deep and thinned ice core from East Antarctica

East Antarctica is a region of particular interest for the drilling of deep ice cores. Indeed, thanks to low accumulation rates, it likely contains the oldest stratigraphically-undisturbed ice on earth. There is currently a search for very old ice, with ages

potentially dating back one-and-a-half million years (Fischer et al., 2013; Passalacqua et al., 2018). When retrieved, such an ice core will be characterized by the gas trapping of low-accumulation East Antarctic Plateau ice cores. Moreover, the old ice will be located close to the bedrock and therefore will be thinned by strain.

In this section, we estimate the potential differences between atmospheric signals and their measurements in a theoretical one-and-a-half million years old ice core, for methane and carbon dioxide. Here, we consider that the gas trapping occurs under an accumulation rate of $2\,\mathrm{cm\,ie\,yr^{-1}}$. We suppose that the enclosed gases follow the age distribution obtained for Vostok during the DO17 period, established to be representative of East Antarctic glacial smoothing in Section 3.4.1. The presence of layered trapping artifacts is simulated using the model parametrized in Section 3.3.2. Finally, we assume that the thinning of the ice results in a resolution of $10,000\,\mathrm{yr\,m^{-1}}$ (Passalacqua et al., 2018). This implies that a given ice layer at the bottom of the core is 200 times thinner than at the firn-ice transition. This has to be taken into account as analytical techniques have limited depth resolutions below which variations are no longer resolved. In particular, layering artifacts will no longer be visible as individual events in the record. Note that in this section we do not take into the diffusive mixing of gases within the ice matrix, that would further smooth the trace gas records (Bereiter et al., 2014).

### 4.1 Methane alterations

As a first case, we study the alterations of a methane record measured using a CFA system analogous to the one currently used at IGE. For the sake of simplicity, we assume that the atmospheric history recorded in this theoretical one-and-a-half million year-old ice core is similar to the DO15 to 17 events of the last glacial period. Hence, we simply use the WD CFA methane measurements as the atmospheric reference (Rhodes et al., 2015). We first determine the initially enclosed signal using the layering artifacts and smoothing models, and apply a thinning factor of 200. The resulting signal is displayed in light blue in Figure 10. Then, we simulate the process of $CH_4$ measurements by applying the smoothing induced by the CFA system. For this, we use the CFA impulse response derived by Fourteau et al. (2017). The end result is displayed in red in Figure 10. As this is a synthetic signal, we plotted the data on a relative depth scale, whose zero was arbitrarily set in the record. The layering artifacts are no longer visible as they have been smoothed out by the CFA measurement. However, they affect the measured signal due to their asymmetry. This is illustrated by the difference between the red and green curves in Figure 10, that represent the CFA measurements with and without the presence of layering artifacts, respectively. However, even during periods of strong methane variations, the bias specifically due to layering artifacts does not exceed $10\,\mathrm{ppbv}$. It therefore appears that layering artifacts only negligibly affect the measured signal in the case of thinned ice cores. However, the potential impact of layering

artifacts is sensitive to the parameters of the layering model. Notably, increasing the number of anomalous layers increases the amplitude of the biases.

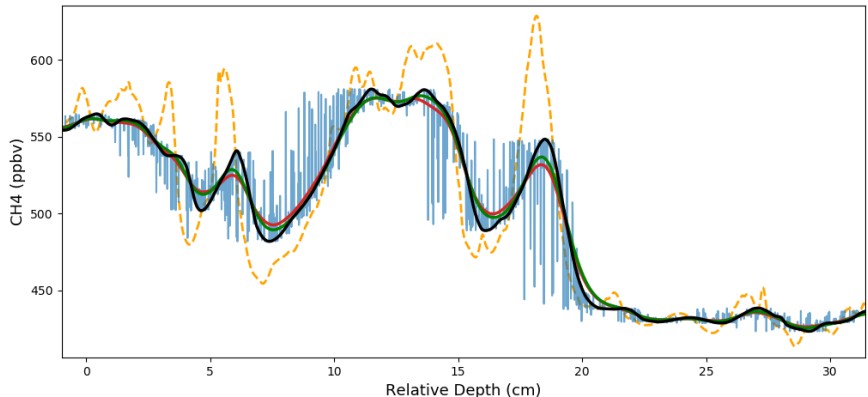

**Figure 10.** Impacts of layered trapping and smoothing on a synthetic million-and-a-half years old methane record. The signal without smoothing or artifacts is displayed in dashed orange. The recorded signal with layering is shown in light blue, and the recorded signal with smoothing only in black. The synthetic CFA measurements taking into account the artifacts are displayed in red, while the synthetic measurements without artifacts are displayed in green. The synthetic depth scale is expressed relative to an arbitrary point of the record. For clarity the depth scale is expressed in centimeters.

The internal smoothing of the CFA system also tends to deteriorate information recorded in the ice core by attenuating fast variability. It is therefore important to determine to what extent CFA smoothing adds to the already present firn smoothing.

To study this point, we compare the frequency responses of the CFA system and firn smoothing. These frequency responses represent the attenuation experienced by a sine signal, as a function of the sine period. The smoothing function of the CFA system (its impulse response) was determined by Fourteau et al. (2017), and originally expressed on a depth scale. To compare it with the firn smoothing, we converted the depth scale of the CFA smoothing function to an age scale by taking into account the accumulation rate and thinning of the ice. Then, the frequency responses were deduced using fast Fourier transforms. The

gas age distribution and the CFA impulse response are displayed in the left panel of Figure 11 with the frequency responses in the right panel. For sine periods longer than $100 \, \text{yr}$ the firn smoothing is larger than the CFA smoothing. Yet, for a sine period of $500 \, \text{yr}$ the firn induces a smoothing of $50\%$ while the CFA system smooths by about $20\%$. The CFA smoothing is therefore noticeable and adds to the already present firn smoothing. This is illustrated in Figure 10, where the difference between the black and green curves is due to the presence of CFA smoothing. The influence of this analytical smoothing would

be even stronger in the case of a narrower gas age distribution, for instance during interglacial periods. This is problematic as it deteriorates climatic information still present in the gas record. It thus appears important to improve the resolution that can

be achieved by current CFA gas systems in order to minimize their internal smoothing, in order to take full advantage of the gas records entrapped in deep ice cores.

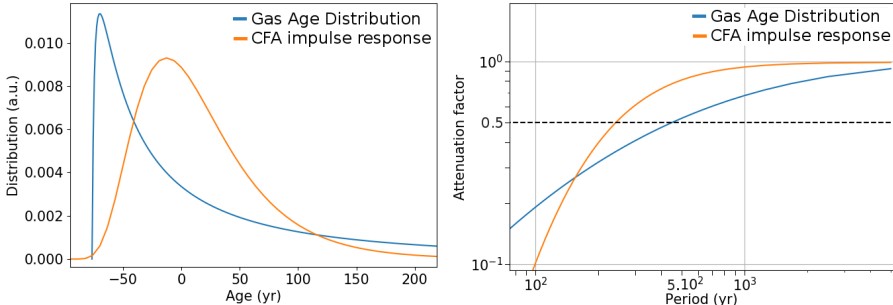

**Figure 11.** Left panel: gas age distribution in the synthetic million-and-a-half years old ice core in blue, and CFA impulse response on a gas age scale in orange. Right panel: attenuation of a sine signal by the firn and CFA smoothing in blue and orange respectively. The horizontal dashed line indicates the $50\%$ attenuation.

## 4.2 Carbon dioxide alterations

So far in this article only methane records have been considered. However, smoothing and layering artifacts also potentially
affect other gaseous records, such as carbon dioxide records. Since $CO_2$ is one of the principal and most studied greenhouse gases, we propose to quantify the potential impacts of gas trapping on the record enclosed in a million-and-a-half years old ice core. Similar to the methane case of the previous section, we used the $CO_2$ measurements of the high accumulation WD ice core as the atmospheric reference (Marcott et al., 2014). The chosen data points correspond to the last deglaciation, as they constitute a fast atmospheric increase of $CO_2$. As for $CH_4$, the synthetic ice core signal is determined with the smoothing and
layering model and ice layers are thinned 200 times. Finally, we emulate the process of discrete measurements. For that, we assume that the final ice core is continuously discretized into $4\,\mathrm{cm}$ thick pieces, and that the measured value is the average value of the concentrations enclosed in the ice piece.

The results are displayed in Figure 12. The ice core signal is displayed in blue in the left panel of Figure 12, with the artifact free record superimposed in black. It can be seen that during enclosure, the layering heterogeneities can produce artifacts reaching
up to $10\,\mathrm{ppmv}$. The measurements obtained by a discrete method are displayed on the right side of Figure 12 with and without the presence of layering artifacts in blue and black respectively. These results show that the impact of layering artifacts on the final measurements does not exceed $0.5\,\mathrm{ppmv}$, even during abrupt $CO_2$ rises, for example at $21$ and $57\,\mathrm{cm}$ relative depth in Figure 12. However, as for methane the impact of the layering artifacts is sensitive to the amount of abnormal layers per meter. We also compared the rate of change of carbon dioxide in the atmospheric reference and in the measured record. For some of

the fast variations of atmospheric $CO_2$, a reduced rate of change is deduced from the ice core record. A specific example (the deeper part of the records in Figure 12) is shown in Figure 13, where the rates of change of $CO_2$ in the atmosphere, recorded in the ice, and deduced from discrete measurements are compared. For this event, the atmospheric rate of change peaks at about $0.055\,\mathrm{ppmv\,yr^{-1}}$, while the corresponding imprint in the ice has only a rate of change of $0.035\,\mathrm{ppmv\,yr^{-1}}$ due to firn

5    smoothing. The process of discrete measurement further diminishes the measured rate of change to about $0.02\,\mathrm{ppmv\,yr^{-1}}$, a bit more than a third of the atmospheric value.

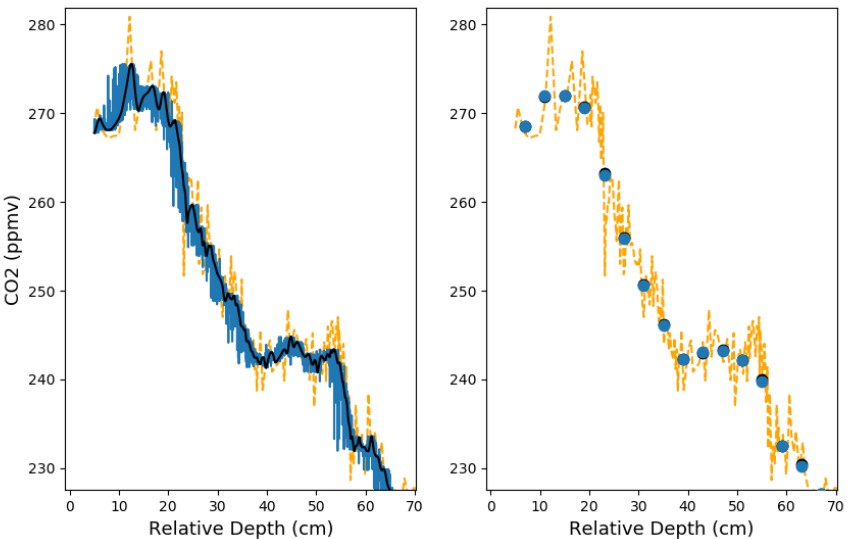

**Figure 12.** Impacts of layered trapping and smoothing on a synthetic million-and-a-half years old carbon dioxide record. Left panel: recorded signal in the ice taking into account smoothing (in black) and layered trapping artifacts (in blue). The signal without smoothing or artifacts is shown in orange. Right panel: measured signal by a discrete system shown as blue points. The signal without smoothing or artifacts (in orange) and the measured signal without layering artifacts (black points) are also displayed. The depth scale is expressed relative to an arbitrary point in the record, and in centimeters.

## 5    Conclusions

This work evaluated two gas trapping effects that affect the recorded atmospheric trace gas history in polar ice cores. The first one is the layered gas trapping, that produces stratigraphic heterogeneities appearing as spurious values in the measured record

10    (Rhodes et al., 2016). The second one is the smoothing effect that removes a part of the variability in the gas records (Spahni et al., 2003; Joos and Spahni, 2008; Köhler et al., 2011). Our work focuses on the arid region of East Antarctica, as this is where the oldest ice cores are retrieved. Five new sections from East Antarctic ice cores have been analyzed at high resolution for methane concentration.

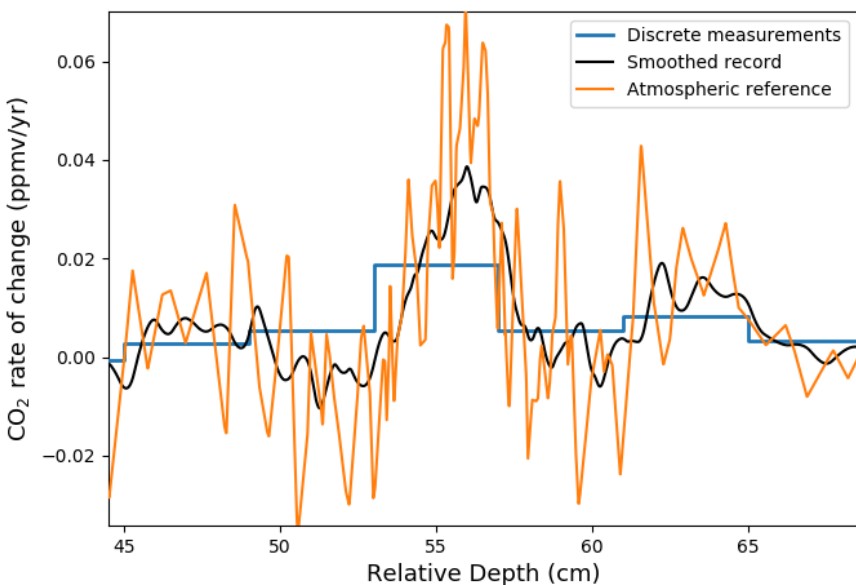

**Figure 13.** Impact of gas trapping processes and measurements on the $CO_2$ rate of change. The rate of change of the atmosphere is displayed in orange, while the rate change recorded in the ice is in black, and the measured one is in blue. The depth scale is the same relative depth scale as in Figure 12.

The characterization of layering artifacts in these new measurements is consistent with the physical mechanisms already proposed in the literature (Etheridge et al., 1992; Rhodes et al., 2016). In accordance with previous studies (Hörhold et al., 2012; Fujita et al., 2016), the data suggest that chemistry could promote the presence of especially dense layers closing in advance and creating the layering artifacts. We parametrized a simple model to reproduce the observed distribution and characteristics

5  of layering artifacts and were able to find a common set of parameters to reproduce the observed distribution of artifacts in all studied East Antarctic ice cores.

In order to constrain the smoothing effect, we estimated the gas age distributions in each of the five ice core sections. For this purpose, we use the method proposed by Fourteau et al. (2017) based on the comparison with higher accumulation methane records. It appears that during the last glacial period, the different methane records in ice cores formed under accumulation rates

10  lower than $2\,\mathrm{cm\,ie\,yr^{-1}}$ all display a similar degree of smoothing. We therefore propose to use a common gas age distribution for glacial East Antarctic sites with accumulations below $2\,\mathrm{cm\,ie\,yr^{-1}}$. The comparison of the glacial and interglacial GADs suggests that the smoothing is higher at Dome C during the glacial period than during the interglacial period. Yet, despite a doubling of accumulation during the interglacial period, the resulting smoothing is only slightly lower. This means that more climatic information than previously thought is preserved in glacial East Antarctica ice core gas records, which are thus suited

to study multi-centennial atmospheric variability. Similarly to Fourteau et al. (2017), we observed that the smoothing during the glacial in the Vostok ice core is weaker than the one predicted for modern Vostok by the age distribution estimated by gas trapping models (Witrant et al., 2012).

Finally, we applied our methodology to the theoretical case of a million-and-a-half years old ice core drilled on the East Antarctic Plateau. Our results suggest that due to thinning, the layering artifacts will no longer be resolved during measurements. However, it appears that their potential influence on the end result measurements is rather low, with impacts below $10\,\mathrm{ppbv}$ and $0.5\,\mathrm{ppmv}$ for methane and carbon dioxide, respectively. In the case of methane variations during DO events, most of the record alterations originate from the firn smoothing. Yet, the limited spatial resolution of the CFA system induces a smoothing that is not entirely negligible compared to the one of the firn. For carbon dioxide, firn smoothing appears to significantly diminish the recorded rates of change of abrupt $CO_2$ increases, compared to their atmospheric values. The estimations of $CO_2$ rates of change are further altered by the process of discrete measurement, and measured values can be three times lower than the actual atmospheric rate of change. Thus under the absence of significant gas diffusion in the ice, very high depth resolution analyses are required for highly thinned ice to resolve centennial $CO_2$ variability to its full extent.

*Code availability.*   The programs used for data processing and modeling were developped using python3 and available packages. They will be provided upon request to the corresponding authors.

*Data availability.*   The high resolution methane datasets will be made available on the World Data Center for Paleoclimatology.

*Author contributions.*   This scientific project was designed by JC, XF, KF, and PM. JC and PM participated in the Lock-In drilling and XF participated in the shallow Dome C drilling. AAE and VL made available and pre-processed the Vostok ice core section. The high resolution methane measurements were carried out by XF and KF. The codes for data processing and modeling were developed by KF and PM. All authors contributed to the interpretation of the data. The manuscript was written by KF with the help of all co-authors.

*Competing interests.*   The authors declare having no competing interests

*Disclaimer.* TEXT

*Acknowledgements.* This work is a contribution to EPICA, a joint European Science Foundation/European Commission scientific program funded by the European Union and national contributions from Belgium, Denmark, France, Germany, Italy, the Netherlands, Norway, Sweden, Switzerland, and the United Kingdom. The Vostok ice core was made accessible in the framework of the Laboratoire International Associé (LIA) Vostok. The Lock-In drilling was supported by the IPEV project No 1153 and the European Community's Seventh Framework Programme under grant agreement No291062 (ERC ICE&LASERS). We are grateful to the Lock-In field personnel: David Colin, Phillipe Dordhain, and Phillipe Possenti who performed the drilling, as well as Patrice Godon for setting up the logistics. The shallow Dome C drilling was performed by Phillipe Possenti and supported by the IPEV project No 902 and the French ANR program RPD COCLICO (ANR-10-RPDOC-002-01). We thank Robert Mulvaney and the British Antarctic Survey for the Fletcher Promontory ice core drilling, and the permission to use the Fletcher methane data. We thank Grégoire Aufresne for his help processing the ice cores. We thank Grégory Teste for his help during the ice core processing and the CFA methane measurements. We thank Jochen Schmitt and Hubertus Fischer for their constructive comments. This work was supported by the French INSU/CNRS LEFE projects NEVE-CLIMAT and HEPIGANE. This work was also supported by the French ANR programs RPD COCLICO (ANR-10-RPDOC-002-01). We are thankful to the two anonymous referees for reviewing this work, as well as to Hubertus Fischer for editing it. This is EPICA publication no. XX and Beyond EPICA - Oldest Ice publication No. YY.

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
