# Peer review of "Estimation of gas record alteration in very low accumulation ice cores"

_Climate of the Past, 2019_

## Referee Comment (RC1) · Anonymous Referee #1 · 19 Sep 2019

General Comments:

The authors present a study of the alteration of atmospheric records preserved in ice cores at Antarctic sites where ice accumulates very slowly. The main point of the paper is to understand how well a measured signal from a low-accumulation ice core reflects the true atmospheric history. The findings are then applied to a hypothetical 1.5 million-year-old ice core record. The topic of this paper is quite specific, but it will be of great interest to those in the ice coring community studying trapped gases and firn processes (particularly using high-resolution instrumentation). The paper will also be of interest to those who seek to develop gas records in ice cores retrieved as part of the oldest ice effort. The authors present a sound study with important conclusions, though the various methods of analysis, data handling, and analytics need to be explained and/

or justified somewhat more clearly in the text to convince readers of their validity (see comments below).

Specifically, the authors investigate the degree of alteration of gas records due to (1) artifacts resulting from early/ late bubble close-off in the firn, and (2) smoothing due to dispersive mixing in the firn column and gradual bubble close-off. The authors present new high-resolution measurements of methane concentration as recorded in five different ice core sections from low-accumulation sites in East Antarctica. To address process (1) the authors identify layer-trapping artifacts in the new datasets including testing a previously published algorithm for artifact identification. The authors also make fundamental observations about the frequency and magnitude of layer-trapping features in the new datasets, and finally they simulate the occurrence of artifacts in a hypothetical 1.5 million-year-old ice core record.

In order to investigate process (2), the authors derive gas age distributions for the new ice core records that reproduce the magnitude of the features resolved in the new data when convolved with an atmospheric reference record. The gas age distributions are explored by applying the smoothing to different datasets, from which the authors come to the conclusion that the gas age distributions of glacial East Antarctic ice cores are very similar. The glacial gas age distribution is then applied to a hypothetical 1.5 million-year-old and highly thinned ice core record to understand the magnitude of smoothing that might be anticipated from such an old ice core.

The data and analyses represent important contributions to the field and are suited for publication in Climate of the Past. However, the article needs significant improvements in clarity and organization for readers to be convinced of the arguments presented therein. Despite the fact that the methods are explained in Fourteau 2017, the derivations of the gas age distributions, the modeling of layer-trapping artifacts, and details about the analytics need more explanation for this article to be standalone. Generally speaking, there are also a number of details related to the analyses that require more justification and/or explanation (see specific comments below).

I recommend the paper is published after major revisions. Please refer to the specific comments below, which I separated into Major Specific Comments, Minor Specific Comments, and Technical Comments (the latter are numerous but very minor, only concerning writing style and typos).

Major Specific Comments:

Introduction – The authors may consider adding a brief description of work that has been done previously to estimate gas record alterations, including information about gas age distributions (what do we expect the distributions to look like? what magnitude of smoothing of gas records has been observed previously?).

Methods – I understand the analytical methods are further described in the Fourteau 2017 paper, but I suggest the authors state at least briefly the analytical precision on the CH4 measurements and any other processing of raw data that occurred. E.g., How are the raw data calibrated? To what standard scale? Are there any corrections besides solubility in the melt stream? I notice that the data are quite noisy (see further comments about the results section), even after cleaning for layer trapping effects – please address this. I also notice there are gaps where data are missing, but the removal of raw data is not explained in the paper. More information about the datasets will help to convince readers that the layer trapping artifacts are real features, not instrumental noise.

Results and Discussion – Readers may be more convinced of the conclusions if the authors (1) explain what they see in the data more exactly and how they judge the results to be "better" or "worse" - here I am referring to the authors' estimations of the gas alteration (see specific comments), (2) explain the limitations of the data more clearly (see specific comments about data noise, data gaps, and analytical noise), (3) more clearly explain how the GADs are derived, and why they behave differently when applied to different ice cores and intervals, and (4) emphasize the main conclusions of the paper more overtly.

A final point that seems missing from this paper would be some discussion about to what extent preexisting, published records may be affected by layer-trapping artifacts or smoothing. It would also be fitting to describe how discrete records might be affected by these processes relative to CFA, given that discrete measurements integrate over multiple annual layers.

Minor Specific Comments:

P2L25 – Can the authors describe these anomalous strata in more detail? How large are they? What is the horizontal/ vertical scale? How/ why do they form?

P3L12 – Please define a closed porosity profile.

P3L13 – Please explain briefly why the closed porosity profiles are associated with larger uncertainties.

P3L29 – In case the reader is less familiar with the topic (and perhaps is not following your rationale), specify that the higher accumulation ice cores have significantly less alteration of the gas records, thus approximating more closely the true atmospheric history.

P4Fig1 – Consider including a second map panel showing the Greenland core mentioned in this paper (NEEM). Also consider adding markers on the map of Antarctica to indicate other ice cores used in the paper (e.g., DE08 and Fletcher).

Methods – A table describing each coring site with local accumulation rate, elevation, coordinates, etc. would be helpful.

P5L4-L21 – Please include the uncertainty on the accumulation rates at each site if available.

P5L18 – Strictly speaking, the CH4 excursions are associated with DO events. I suggest rewording this sentence accordingly.

P6L5 – Perhaps say why the solubility correction might change between ice cores.

P6L23 – It would be helpful to see a supplementary figure showing the agreement between the raw data and the records used for correction (WD and EDC). Earlier the authors stated that the solubility correction could be different between cores, but now they state the data are all corrected by about +13%. So there are no differences in solubility between the different cores? Could the solubility change due to changes in the dust or ion content in the ice, i.e. during an interstadial versus a stadial? Could this affect the estimation of the smoothing?

P6L23 – 1.13 is a rather high solubility correction. Can the authors say why, or perhaps compare to corrections used by other groups?

P7L5 – Is it problematic that the atmospheric reference dataset changes below 330 m? It seems like this point deserves another sentence of explanation or justification.

P7L8 – Is the gas age scale for Fletcher Promontory also consistent with AICC 2012? Please address how the age scale uncertainties affect your results, given that the records used in your analysis are sometimes on different age scales.

P7L13 – Have the authors considered using EDML, Taylor Dome, or Talos Dome data to estimate or confirm the gradient you determined? These cores may not necessarily fit the criteria of "weakly smoothed" and "high-resolution," but they could still be useful. If not, at least mention them and state why the authors didn't use them.

P7L14 – What spline version of the NEEM data do the authors refer to? The data I have seen from NEEM have many gaps due to instrumentation problems, including the CH4 peak associated with DO 21. Did the authors fit the splines themselves? If so, please explain.

P7L14-L15 - It seems like this statement about the gas age distribution comes too early. The reader does not yet know how the authors determine the gas age distributions, so they cannot (at least at this point) easily see for themselves how the authors have come to this conclusion. In any case, doesn't the need to deconvolve the NEEM dataset imply

that NEEM firn has smoothed the true atmospheric signal? Is the implied smoothing at NEEM reasonable?

P7L18 – How much does the deconvolution increase the amplitude of the DO21 methane event in NEEM?

P7L20 – Can the authors estimate the extra bias/ uncertainty introduced by the deconvolution of the NEEM record, including using the NEEM gas age distributions as input? Or at least justify that it does not significantly affect the conclusions.

Section 3.2 – A table showing which atmospheric reference is paired with which new dataset would be helpful. Additional information such as the age scale of the atmospheric reference dataset, the time interval, the measurement technique used, the resolution of the data, the accumulation rate at the core site, and any treatments to the reference dataset (e.g., spline fit, deconvolution) could also be listed.

Section 3.2 - A few more thoughts: (1) Could NGRIP be used instead of NEEM? Would the NGRIP record also need to be deconvolved to be consistent with Vostok at DO21? (2) Can the authors say more about why there is a need to deconvolve the NEEM signal while other reference datasets appear to work without the deconvolution? (3) I can't help but notice there are actually not any new CFA data at 1260 m in Vostok. There is a large data gap between 1260-1261 m or so. So the authors are assuming the height of the CH4 feature there? How did they do this?

P8Fig2 – Legends on this figure and others would be very helpful. Please label the events in the figures (e.g., DO6, DO7, DO8, etc.). Why are there data gaps in the new CFA records? For this figure and others that follow, plotting the atmospheric reference on depth is very confusing. The depths are obviously not the same in the reference core as for the new data – they are from different cores.

P8Fig3 – See comments about Figure 2 with respect to plotting the reference records on depth.

P9Fig5 – For this figure and for others, the reader must follow multiple cross-references to different captions in order to understand what is being plotted. Consider adding legends to the figures for clarity, or writing full captions for each figure rather than cross-referencing.

P9Fig5 - Please explain how one can measure a layer-trapping artifact that is higher in CH4 concentration than any measured value during DO8? I'm referring to the highest light blue value in Figure 5 near 740 m.

P9L2 – This is the first mention of the analytical noise. Consider putting this in the methods section where the authors discuss the analytical technique.

P10L5-8 - This is a very fundamental and robust conclusion – that late close-off artifacts are rare relative to early close-off artifacts. What does that mean for the GADs? Is there a connection?

P11L3 – What about Na+ or other ions? How good is the assumption that Ca2+ is a predictor of total ion content?

P11L6 – How many volcanic markers are there common to both cores in the depth intervals relevant here?

P11L10 – Exactly how are the layer trapping artifacts identified? This should be explained more clearly, possibly in the methods section. How can the authors be sure these are not analytical artifacts?

P11L20 – Do the DO17 data also show a high number of layer artifacts?

P11L26 – What do the authors mean by the "bulk behavior?" How is that defined?

P12L14 – How sensitive is the model to the assumed densification rate?

P13L6 – I don't see a "clear underestimation" of the layering artifacts for DO events 7 and 8 in Figure S4. Please clarify what the authors see in the data.

It would be helpful to also plot the reference records in the upper panels of these plots so readers can compare how the data look relative to the better record of atmospheric history.

P13L9 – I don't understand which visual results the authors are judging to be correct here, so the 1.5 factor seems arbitrary.

P14L3 – How is the analytical noise estimated?

P14L5 – Here (in the Holocene data) I can't help but question if the authors have actually identified real layer trapping artifacts or just the outlier analytical noise. Can they really distinguish? Please provide some justification.

P14L11 – Please justify why providing a manually specified artifact-free signal is not circular. If the authors can visually identify the artifacts so straightforwardly, why then is it necessary to run the algorithm at all?

P17L6-10 – This seems like the main point of this paragraph and possibly one of the more important conclusions of the paper. Consider starting the paragraph with this sentence and filling in the details thereafter.

P17Fig10 – I'm still having trouble understanding how the atmospheric references can be plotted on the depth scale of a different ice core. Also consider plotting the atmospheric reference in dashed orange lines to be consistent with previous figures.

P17Fig10 - I would like to see more discussion of why the various GADs estimate the smoothing of DO6-9 similarly well, but they have larger discrepancies for DO21.

P18Fig11 – Again the reference is plotted on depth, which is very confusing without further explanation. Consider plotting in dashed orange to be consistent, similar to previous comment.

P19L2 – How does the skew relate to the firn? What's the physical reason for the skew?

P19L8 – Can the authors explain this conclusion further? Readers may think synchronizing to a high-accumulation record would increase the accuracy of the gas chronology.

P21Fig12 – The artificial depth scale is confusing, can you explain where it comes from?

PS1Fig1 - Again the NEEM reference data are plotted on the same depth scale as the Vostok DO21 data, which is confusing.

PS1Fig1 - I think a problem with the point the authors are trying to make here is that there is a large data gap at the onset of DO 21. The full magnitude of the CH4 rise in the Vostok DO21 record is not technically resolved.

Technical Corrections:

The following are suggestions to improve the writing and readability of the paper. They are mainly word changes, clarifications, and minor grammar mistakes.

P1L1 – "East Antarctic Plateau" rather than "East Antarctic plateau" This is throughout the paper.

P1L2 – Change "affect" to "influence" or a word that is not so similar to the preceding word "effects" . P1L6 – "Concentration measurements..., which removes..." Do the authors want to say the concentration measurements themselves alias the fast variability? Or that the fast variability is removed because of smoothing in the firn, as a consequence of layers containing gas with a distribution of ages? Reword.

P1L15 – Change "estimate" to a word that is less similar to the preceding word "estimation."

P1L15 – Change to "...their potential impacts on a hypothetical million-and-a-half years old ice core..."

P1L18 – Change to "...in the case of methane and carbon dioxide, respectively."

P2L2 – I prefer "accumulation" rather than "precipitation" because accumulation is the net sum of precipitation and removal by sublimation and wind scouring. Also "precipitations" should be singular.

P2L4 – Remove "back" so it reads "dated to 800,000 years..."

P2L6 – Change to "...the reconstruction of Earth's past temperatures..."

P2L14 – Change to "...in low accumulation ice cores cannot be interpreted as a perfect record of the atmospheric history."

P2L15 – Remove "might."

P2L16 – Add "...imprint in the ice."

P2L26 – Remove the comma.

P3L4 – Change to "...degree of alteration between the actual atmospheric history and the signal recorded in ice cores is strongly dependent..."

P3L10 – Change to "Gas age distributions can be calculated for the purpose of estimating firn smoothing, and in the case of modern ice cores this may be accomplished by using gas trapping models parameterized by firn air and pore closure data (refs)."

P3L15 – Reword "...temperatures that have no known equivalent nowadays" to be more specific about what the authors mean. Also "nowadays" sounds like slang, even though it is technically a real word.

P3L16 – Change to "Thus it is not possible to sufficiently constrain gas trapping models, prohibiting robust estimation of the gas age distributions that were responsible for smoothing during glacial periods."

P3L17 – Consider starting a new paragraph at "Concerning..." Also change the wording to "Layered gas trapping, on the other hand, originates from firn heterogeneities and is a stochastic process."

P3L26 – The authors spell "parameterize" differently here. Pick one spelling to be consistent throughout the paper.

P3L26 – "gas layered trapping" or "layered gas trapping?"

P3L26 – Combine the two sentences to read "...proposed by Fourteau et al. (2017), with the goal of rendering the model applicable..."

P4L4 – Add to the end "...the deterioration of atmospheric information due to firn smoothing and layered gas trapping."

P4L13 – Change to "The local accumulation is 3.9cm.ie.yr-1 (Yeung et al. 2019)."

P4L15 – Change to "Firn air sampling during..."

P5L1 – 1950 CE or AD 1950.

P6L2 – 3.6cm.ie.yr-1 can't be the right unit, you would melt a meter of ice in 28 years.

P6L2 – Change wording to "..., which resolves centimeter scale variations in the methane record."

P6L3 – I don't think the authors mean to say "preferential" here. CH4 is less soluble in water than other greenhouse gases measured in ice cores. This term is used in other parts of the paper, too.

P6L4 – Change to "It is therefore necessary to apply a correction factor to account for the solubility effect."

P6L5 – Change to "and potentially differs between ice cores."

P6L5 – Change to "...methane dissolution is addressed in Section 3.1 below."

P6L21 – Change to "The construction of the gas age chronologies indicated in the figures is described in Section 3.4.1."

P6L26 – Change wording to "For the atmospheric references we used methane gas

records from higher-accumulation ice cores where higher frequency variations are preserved relative to the low accumulation ice cores."

P7L4 – "Fletcher Promontory" rather than "Fletcher promontory"

P7L12 – Change to "...possible that the inter-hemispheric gradient was not constant..."

P7L15 – Change "all" to "full" or "complete."

P8Fig2 – Change the caption to read, "The blue and yellow spikes are randomly distributed early and late closure artifacts, respectively."

P9L2 – Change to "during periods when the atmospheric CH4 concentration was varying."

P9L4 – Add a comma after "respectively."

P10L7-8 – Change "long distance gas transport" to "gas transport to greater depths." Change "...late closer layers from enclosing young air (ref)."

P11L13 – Change "in advance" to "relatively shallower in the firn."

P11L22 – Change "quantifying" and "predicting" to "quantify" and "predict."

P11L26 – Change to "...its density difference relative to the bulk behavior."

P12L5 – "The Fourteau et al. (2017) model..."

P12L12 – The authors might consider writing "2.2kg.m-4" as "2.2kg.m-3m-1."

P13L8 – Change to "...during periods of high calcium..."

P13L11 – Change to "...displays calcium variability similar to..."

P14L7 – Add commas between core names.

P14L9 – Use a different word besides "concentration" to describe the number of artifacts. One might confuse it with CH4 concentration.

P15L4 – Use a different word besides "important." Perhaps "significant" or "potentially significant."

P15L13 – Consider using the word "function" instead of "law."

P15L14 – Change "in the mathematical sense" to "in a mathematical sense."

P15L14 – "independent" instead of "independents."

P15L17 – "new gas age chronologies" instead of "a new gas age chronologies."

P16L2 – Change to "Low-temperature ice deforms less easily, slowing the densification process such that bubble closure spans a larger time period."

P16L2 – Remove "a" so it reads, "Moreover, low accumulation is also. . ."

P17L7 – Change "exhibits" to "exhibit" and "degree" to "degrees."

P18L2 – "Latter" instead of "later." Also add "is" to "This is illustrated with. . ."

P18L4 – Use different wording than "more important." Perhaps "more significant" or simply "greater."

P18L12 - Needs a figure reference.

P18L13 – Instead of "less variability" consider using "smaller magnitude." Readers may confuse the term variability to mean noise or variation around a mean value.

P19L23 - I think the authors mean "depth resolution" instead of "spatial resolution."

P20L6 – Change to "apply a thinning factor of 200."

P20L23 – This is not easy to see on Figure 13. Consider labeling more of the axis tick marks and increasing the font size.

PS1L2 – Change the word order to "In the main article, we used the deconvolved

methane CFA data of Chappellaz et al. (2013) as atmospheric reference for the Do21 period.

PS1L3 – Change to "This was done because using the data..."

PS2L2 – "Figure 1 of Fourteau et al. (2017)."

PS2Fig2 – State briefly how the data are cleaned for layering artifacts in the caption.

PS2L7 – Change to "...pointed out that one of the data points of Loulergue et al. (2008) might correspond to an early closure artifact and may not be climatically relevant."

PS5FigS7-FigS10 – Same as Figure "S6"
* * *

---

## Referee Comment (RC2) · Anonymous Referee #2 · 27 Sep 2019

In the next few years, new data from Antarctic low accumulation sites will push the ice core records to 1.5 million years or older. The annual layer thickness in these ice cores is very small, not within reach of the measurement resolution. The resulting gas record will therefore be smoothed. The processes addressed in this manuscript alter the (naturally) smoothed trace gas record. Understanding the modulation of the natural record is needed and of high importance. The paper is certainly suitable for Climate of the Past. However, I find the manuscript lacks clarity and needs revision. Specifically the method needs better explanation. The entire manuscript needs to be reworked and may benefit from reorganization and shortening. Major revisions are necessary before it is publishable.

P3, Line15: Please write the units out the first time. 2 cm ice equivalent yr-1 (cm ie

yr-1). No dot between cm and ie; otherwise it means cm times ice equivalent.

Please make it clear early in the manuscript that you prefer to use Antarctic high resolution data for comparison as they are not affected by the pole to pole gradient.

There are two continuous records from NEEM. Explain why you prefer to take the one you do or explain that it does not matter, does it?

I have a feeling on how the model works but the mathematical formulation on page 12 does not make sense. Unit wise that equation is definitely wrong. The model needs to be explained in depth and better before this manuscript is publishable.

The manuscript has too many figures where records are also unnecessarily repeated. I suggest fewer graphs. The graphs also lack information on which record the Antarctic data is compared to. Please label Dansgaard-Oeschger events in the manuscript. When sections of the core are compared in the text, it would help to have them labeled in the graph.

Supplemental: S1 The depth scale seems to apply to Vostok not to NEEM. What section of the NEEM core is that? It is quite obvious that there is a gap in the original record that leads to the too much smoothed record. The conclusion about NEEM gas age is not supported in my opinion.

---

## Author Comment (AC1) · 15 Nov 2019

**Response to Referee #1 – cp-2019-94**

We are thankful to the referee for their constructive comments on the article.
We listed below our responses to the major and minor specific comments. For clarity we have sometimes regrouped different comments of the referee, to provide a common answer.
The comments of the referee are in blue, and our corresponding responses are below in black.

We did not directly answer the technical comments, but took them into account.

Kévin Fourteau on behalf of all co-authors

MAJOR SPECIFIC COMMENTS:

Introduction – The authors may consider adding a brief description of work that has been done previously to estimate gas record alterations, including information about gas age distributions (what do we expect the distributions to look like? what magnitude of smoothing of gas records has been observed previously?).
We will further describe the work already performed on layering artifacts and smoothing in East Antarctica ice cores **P3L7**:
*"For layered gas trapping, Fourteau et al. (2017) report artifacts reaching up to 50ppbv in the Vostok methane record during the Dansgaard-Oeschger event 17 period. For smoothing, Sphani et al. (2003) report a gas age distribution in the EPICA Dome C ice core dampening atmospheric variability faster than a few hundred years. Similar degrees of smoothing have also been reported by Köhler et al. (2015) and Fourteau et al. (2017) for the ice cores of Dome C and Vostok, respectively."*

Methods – I understand the analytical methods are further described in the Fourteau 2017 paper, but I suggest the authors state at least briefly the analytical precision on the CH4 measurements and any other processing of raw data that occurred. E.g., How are the raw data calibrated? To what standard scale? Are there any corrections besides solubility in the melt stream? I notice that the data are quite noisy (see further comments about the results section), even after cleaning for layer trapping effects – please address this. I also notice there are gaps where data are missing, but the removal of raw data is not explained in the paper. More information about the datasets will help to convince readers that the layer trapping artifacts are real features, not instrumental noise.
We will rewrite the high-resolution measurements section with more detail:
*"The five ice core sections were analyzed for methane concentrations using a Continuous Flow Analysis (CFA) system, including a laser spectrometer based on optical-feedback cavity enhanced absorption spectroscopy (OF-CEAS; Morville et al., 2005), at the Institut des Géosciences de l'Environnement (IGE), Grenoble, France. The laser spectrometer was calibrated to the NOA2004 scale (Dlugokencky et al., 2005) using three synthetic air standards of known concentrations. The five ice core sections were melted at an average rate of 3.6cm/min, which resolves centimeter scale variations in the methane record (Fourteau et al., 2017). Yet, the measured concentrations are affected by the preferential dissolution of methane compared to nitrogen and oxygen in the meltwater (Chappellaz et al., 2013; Rhodes et al., 2013). It is therefore necessary to apply a correction factor to account for the solubility effect. However, this factor is a priori not known and potentially differs between ice core measurement campaigns depending on factors such as the air content of the ice, or the precise CFA set-up. The methodology for correcting for methane dissolution is addressed in Section 3.1 below. Using a mixture of de-ionized water and standard gases, the analytical noise of the CFA system has been determined to be about 10ppbv peak-to-peak*

*(Fourteau et al., 2017).*
*The obtained records present numerous gaps, ranging from a few centimeters to several meters. Several reasons explain the presence of such gaps. First, the space between consecutive melting sticks let modern air enter the CFA system, resulting in a contamination and abnormally high methane concentrations. Moreover, the presence of cracks and fracture in the ice might also let modern air enter the measured ice stick itself, also resulting in abnormally high concentrations. The moments of potential air intrusions were recorded during the measurement campaigns, and the data were screened to remove the resulting contaminations, creating gaps in the record (Fourteau et al., 2017). Finally, some of the ice was simply not available for this study, resulting in further gaps in the records. This notably explains the gap visible around the 1260m depth in the Vostok record (Figure 6)"*

Results and Discussion – Readers may be more convinced of the conclusions if the authors (1) explain what they see in the data more exactly and how they judge the results to be "better" or "worse" - here I am referring to the authors' estimations of the gas alteration (see specific comments), (2) explain the limitations of the data more clearly (see specific comments about data noise, data gaps, and analytical noise), (3) more clearly explain how the GADs are derived, and why they behave differently when applied to different ice cores and intervals, and (4) emphasize the main conclusions of the paper more overtly.

(1): To simplify and clarify the article we have decided to remove the calcium parametrization of layering artifacts model. At this point, our results are probably not robust enough to be presented in the paper.
We will clarify how we judge the performance of the layering artifacts model **P13L15**:
*"The model produces layering artifacts in the expected parts of the records (mostly right after the onsets of DO events) and with the expected sign and amplitude. This is visible on Figures 2 to 6, where the layering artifacts of the records fall within the expected envelopes of the model results."*

(2): We will clarify how we determine the presence of layering artifacts in the record, and why they cannot be explained by noise or contamination issues **P9L1**:
*"In accordance with Rhodes et al. (2016) and Fourteau et al. (2017), we observe abrupt centimeter-scale variations in the records during periods when the atmospheric $CH_4$ concentration was varying, that we interpret as layering artifacts. We identified such layering artifacts as spikes with widths of a few centimeters and whose concentrations are larger than the analytical noise of ~10ppbv. Despite the effort to clean the record of modern air contamination, it is possible that some contamination spikes remain and would then be wrongly interpreted as layering artifacts. However, the presence of negatively orientated spikes that cannot be attributed to contamination, and the overall repartition of the spikes during periods of fast atmospheric methane variation confirm that these centimeter-scale abrupt variations are mostly due to the mechanism of layered gas trapping."*

We will add a comment on the impact of the gaps in the records on the extraction of GAD **P19L6**:
*"The GAD extraction procedure is designed to match the CFA record and the convoluted atmospheric reference only where CFA data are available. We thus do not have to extrapolate the CFA data within the gaps of the record. However, the absence of gaps in the records would have reduced the uncertainty of the estimated GAD, as the extra data would have provided more constraints on the GAD estimation."*

(3): We will rewrite the description of the method with more details **P15L10**:
*"This method, designed to estimate the GAD of low accumulation records, is based on the comparison with a weakly smoothed record derived from a high accumulation ice core, used as an input atmospheric reference (see Section 3.2). The idea of the method is to find a GAD that is able to smooth the atmospheric reference into the low-accumulation record. We thus searched for the*

*GAD that minimizes the RMSD between the CFA measurements and the smoothed version of the atmospheric reference. In order to have a well-defined problem in a mathematical sense, the GAD of the low-accumulation ice core is assumed to be a log-normal function. Such a log-normal distribution is fully defined by two independent parameters. Finding the best GAD to smooth the atmospheric reference is then reduced to the recovery of a pair of optimal parameters. Nonetheless, log-normal distributions exhibit a large range of shapes that can adequately represent age distributions (Köhler et al., 2011; Fourteau et al., 2017). In order for the GAD extraction to perform well, it is also necessary to have a well defined gas age chronology for the low-accumulation record, so that its methane variability is well aligned with the atmospheric reference. This chronology is built recursively during the GAD extraction procedure by manually selecting tie-points, in order to be consistent with the already-existing chronology of the atmospheric reference (Fourteau et al., 2017)."*

We will add a paragraph commenting why the DO6-9 and DO21 periods behave differently when smoothed **P17L3**:
*"However, the uncertainty analysis in Section S5 of the Supplement reveals that the age distribution extracted from the EDC96 record is poorly constrained. This indicates that the smoothing of DO6-9 period is not sensitive to the choice of GAD, and that a large range of GADs results in adequate smoothing for the EDC96 record. On the other hand, the DO21 period is very sensitive to the choice of GAD, and fewer age distributions are able to reproduce the smoothing of the Vostok DO21 record. Because of its shape and its fast atmospheric rate of change (Chappellaz et al., 2013), the first feature of DO21 is sensitive to the choice of GAD, despite a gap in the record, and is therefore a good discriminant between potential age distributions. On the other hand, the step-like features of the DO6-9 record are less sensitive to the choice of GAD, which leads to a less-well constrained extraction of age distributions. Consequently, the distributions obtained for glacial Vostok (DO17 and 21 events) are also suited for the smoothing of DO6 to 9 events in EDC96, as seen in the lower panel of Figure 10."*

(4): We will add to the start of the paragraph **P16L6**:
*"Our results suggest that, during the glacial period, East Antarctica ice cores are affected by the same level of smoothing, that can be represented with the same gas age distribution ."*

We will also add to the start of the paragraph **P18L1**:
"Our results also indicate that the glacial and inter-glacial smoothings of East Antarctica ice cores are relatively similar. Indeed, [...]"

We will also add in the conclusion **P24L13**:
*"This means that more climatic information than previously thought is preserved in East Antarctica ice core gas records, which are thus suited to study multi-centennial atmospheric variability."*

A final point that seems missing from this paper would be some discussion about to what extent preexisting, published records may be affected by layer-trapping artifacts or smoothing. It would also be fitting to describe how discrete records might be affected by these processes relative to CFA, given that discrete measurements integrate over multiple annual layers.
We will add more information of the discrete measurements of layering artifacts and the Loulergue et al. (2008) data in particular **P9L8**:
*"The usage of high-resolution measurements allow us to easily distinguish layering artifacts as abrupt spikes exceeding the analytical noise in the record. However, with a lower-resolution technique, such as discrete measurements, it is possible to inadvertently measure a layering artifact without realizing that it is not representative of its surrounding concentrations, which would result in an anomalous point in the record. As a particular example, Fourteau et al. (2017) pointed out*

*that one of the data points of the DO8 event in the EPICA Dome C methane record published by Loulergue et al. (2008) might have been sampled in an early closure layer, and thus represents a spurious value in their record. By comparing our new high-resolution record with the one of Loulergue et al. (2008), we confirm that this data point does not correspond to atmospheric variability, and that it was sampled in a zone with a high number of early closure artifacts (see Figure~S5 of the Supplement). In the case where such an abrupt variation is observed with discrete measurements, an additional sample should be measured in the vicinity of the first sample in order to confirm that the variability is not due to layered gas trapping."*

MINOR SPECIFIC COMMENTS:

P2L25 – Can the authors describe these anomalous strata in more detail? How large are they? What is the horizontal/ vertical scale? How/ why do they form?
These anomalous strata are a few centimeter thick layers. However, we do not know their horizontal extension, as our observations are limited to the diameter of firn cores. The specific origin of these early/late closure strata is still an open question, although they appear to correspond to strata with density anomalies (Fourteau et al., 2019, In Press).

We will modify the sentence to mention density **P2L20**:
*"However, firn is a highly stratified medium (Freitag et al., 2004; Fujita et al., 2009; Hörhold et al., 2012; Gregory et al., 2014) and some especially dense strata (respectively less dense strata) might experience early (respectively late) pore closure when compared to the rest of the firn (Etheridge et al., 1992; Martinerie et al., 1992; Fourteau et al., 2019)."*

P3L12 – Please define a closed porosity profile.
We will modify the article **P3L12** with:
*"However, to estimate the gas age distribution in bubbles it is necessary to use a depth-profile of the progressive closure of pores in the firn, quantifying the transformation of open pores into closed bubbles."*

P3L13 – Please explain briefly why the closed porosity profiles are associated with larger uncertainties.
We will modify the sentence **P3L13** to:
*"Due to the re-opening of closed pores at the surface of the firn samples used for porosity measurements, such profiles of pore closure are associated with large measurement uncertainties (Schaller et al., 2017, Fourteau et al., 2019)."*

P3L29 – In case the reader is less familiar with the topic (and perhaps is not following your rationale), specify that the higher accumulation ice cores have significantly less alteration of the gas records, thus approximating more closely the true atmospheric history.
We will add the sentence **P3L29**:
*"Indeed, gas trapping occurs faster at high-accumulation sites, which are thus less affected by both smoothing and layering artifacts and can therefore be used to produce atmospheric scenarios with low levels of alteration."*

P4Fig1 – Consider including a second map panel showing the Greenland core mentioned in this paper (NEEM). Also consider adding markers on the map of Antarctica to indicate other ice cores used in the paper (e.g., DE08 and Fletcher).
We will add a map showing the location of both the Greenland and Antarctic sites

Methods – A table describing each coring site with local accumulation rate, elevation,

coordinates, etc. would be helpful.
P5L4-L21 – Please include the uncertainty on the accumulation rates at each site if available.
We will add a table describing the sites, their location, and their accumulation rates.

The AICC chronology do not provide uncertainties for the accumulation rate. We will nonetheless provide the variability (as standard deviation) of the accumulation over the studied sections.
Note that as the last glaciation is recorded in the upper part of the Vostok and Dome C cores, the thinning rate is small and the accumulation rates are thus well constrained by the chronological synchronization with North GRIP.

P5L18 – Strictly speaking, the CH4 excursions are associated with DO events. I suggest rewording this sentence accordingly.
We will modify the sentence to:
*"The CH4 excursion associated with the Dansgaard-Oeschger (DO) events 6 to 9 are included in this gas record (Huber et al., 2006; Chappellaz et al., 2013)."*
We will also modify **P5L23**:
*"This section was chosen as it includes the record of the CH4 excursion associated with the DO21 event, the fastest methane increase of the last glacial period (Chappellaz et al., 2013)."*

P6L5 – Perhaps say why the solubility correction might change between ice cores.
We will modify the sentence to:
*"However, this factor is a priori not known and potentially differs between ice core measurement campaigns depending on factors such as the air content of the ice, or the precise CFA set-up."*

P6L23 – It would be helpful to see a supplementary figure showing the agreement between the raw data and the records used for correction (WD and EDC). Earlier the authors stated that the solubility correction could be different between cores, but now they state the data are all corrected by about +13%. So there are no differences in solubility between the different cores? Could the solubility change due to changes in the dust or ion content in the ice, i.e. during an interstadial versus a stadial? Could this affect the estimation of the smoothing?
P6L23 – 1.13 is a rather high solubility correction. Can the authors say why, or perhaps compare to corrections used by other groups?
The dissolution coefficient found for this study varies between 1.12 and 1.14. This will be written clearly in the text.
As these factors are obtained by comparing our data to already calibrated data, it is not clear whether these variations of dissolution factors are indicative of actual differences in dissolution, or just reflect small inconsistency between the different calibration data.

Changes in dissolution could either be due to a large change in air content (between high and low-accumulation ice core for instance), or a change in the CFA set-up (for instance changing the geometry of the melthead). For instance, measuring the NEEM ice core at with the Desert Research Institute, Reno, US, CFA system lead to a dissolution of 1.079.

The 1.12 to 1.14 reported in the article are consistent with the 1.125 value that we measured for the Vostok ice core during the DO17 period (Fourteau et al., 2017).

We have no reason to think that there should be a difference in dissolution between stadial and interstadial periods, as there were no major change in climatic conditions at Antarctic sites, and therefore no expected variations of air content. Moreover, our data do not suggest a variation in dissolution between stadial and interstadials periods.

We will modify **P6L23** with

*"The correction factors range between 1.12 and 1.14. They are close to the value of 1.125 reported by Fourteau et al. (2017) for the Vostok ice core with the same IGE CFA system, but larger than the 1.079 correction factor reported by Rhodes et al. (2013) for the NEEM ice core with a different CFA system. This difference could to be due to the larger air content in NEEM as well as the different CFA systems, including the melthead geometry."*

P7L5 – Is it problematic that the atmospheric reference dataset changes below 330 m?
It seems like this point deserves another sentence of explanation or justification.
It is not problematic as the lower part of the dataset does not display fast atmospheric variations to be smoothed. Therefore, using a lower accumulation record do not deteriorate the quality of the methane composite for our application. We will add the sentence **P7L5**:
*" Using the low-accumulation EDC99 record for the lowest part does not deteriorate the quality of the composite for our application, as this part does not include fast atmospheric variation to be smoothed, and using a low-accumulation record is therefore appropriate in this case."*

P7L8 – Is the gas age scale for Fletcher Promontory also consistent with AICC 2012?
Please address how the age scale uncertainties affect your results, given that the
records used in your analysis are sometimes on different age scales.
The Fletcher Promontory record was synchronized with the WAIS Divide ice core with manual tie points. We will add the sentence **P7L6**:
*"The dating of the Fletcher-EDC99 composite was made consistent with the WAIS Divide chronology by manually selecting tie points (Buizert et al., 2015)"*

P7L13 – Have the authors considered using EDML, Taylor Dome, or Talos Dome data
to estimate or confirm the gradient you determined? These cores may not necessarily
fit the criteria of "weakly smoothed" and "high-resolution," but they could still be useful.
If not, at least mention them and state why the authors didn't use them.
Using EDML to estimate the inter-hemispheric gradient yields a value between 25 and 35ppbv, which is consistent with the 30ppbv used in the article.

For the article, our goal is not to provide a robust inter-hemispheric gradient, but to ensure that the Vostok and NEEM records are as consistent as possible. It will be made clearer in the text **P7L11**:
*"As the Vostok and NEEM sites are located in different hemispheres, it is necessary to take into account the inter-hemispheric methane gradient between the two sites to make the two records as consistent as possible. Using the Vostok and NEEM records, we evaluated this inter-hemispheric gradient to be 30ppbv for the DO21 period. This value is in line with the work of Dallenbach et al. (2000), and has been corroborated using the EDML methane record (EPICA Community Members 2006).*

P7L14 – What spline version of the NEEM data do the authors refer to? The data I
have seen from NEEM have many gaps due to instrumentation problems, including the
CH4 peak associated with DO 21. Did the authors fit the splines themselves? If so,
please explain.
We performed the splining of the NEEM data in order to fill the missing data of the signal. It will be added in the text **P7L14** and shown in the Supplement:
*"The original data of Chappellaz et al. (2013) present various gaps, that were filled by computing a splined version of the original NEEM CFA record. The spline was chosen not to induce smoothing in the NEEM record and to overlap with the original dataset in parts where data already existed. It is shown together with the original Chappellaz et al. (2013) data in Figure S2 of the Supplement."*

DECONVOLUTION OF NEEM
P7L14-L15 - It seems like this statement about the gas age distribution comes too early. The reader does not yet know how the authors determine the gas age distributions, so they cannot (at least at this point) easily see for themselves how the authors have come to this conclusion. In any case, doesn't the need to deconvolve the NEEM dataset imply that NEEM firn has smoothed the true atmospheric signal? Is the implied smoothing at NEEM reasonable?
P7L18 – How much does the deconvolution increase the amplitude of the DO21 methane event in NEEM?
P7L20 – Can the authors estimate the extra bias/ uncertainty introduced by the deconvolution of the NEEM record, including using the NEEM gas age distributions as input? Or at least justify that it does not significantly affect the conclusions.

Yes our understanding is that the NEEM signal is too much affected by firn smoothing to be used as the atmospheric scenario during the DO21 period for the Vostok ice core. In particular the first feature of the D021 is partially smoothed in the NEEM record, which means that is no longer representative of the atmospheric variability.
The deconvolution of NEEM is necessary to retrieve the amplitude of the first feature of the DO21. In order to do it we assumed that the DO21 GAD of NEEM is similar to the modern GAD of Siple Dome (Witrant et al., 2012), since they share a similar accumulation.
We will show in the Supplement the different deconvolution, in comparison with the non-deconvoluted data.

To test the sensitivity of the deconvolution to the choice of the NEEM GAD, we performed several deconvolution with GADs corresponding to the high (Berkern Island) and low-end (South Pole) of the potential NEEM accumulation range. It indicates that the three deconvolutions produce similar atmospheric references. Moreover, the differences are almost entirely removed when the atmospheric references are smoothed with a glacial East Antarctica age distribution.

We have also performed the GAD extraction procedure for each of the three deconvolutions. The resulting GADs are similar to each other, and fall within the uncertainty range provided in the sensitivity analysis Section S5 of the Supplement.

We will rewrite the end the atmospheric reference section, starting **P7L14**:
"*Yet, we observed that the NEEM record cannot be directly used as an atmospheric input for the DO21 period. As explained in Section S2 of the Supplement, the first feature of the DO21 event (around the 1260m depth in Figure 6) is partially smoothed in the NEEM record. To retrieve the full amplitude of this fast event, we used the deconvolution technique described in Witrant et al. (2013) and Yeung et al. (2019). To be applicable this method needs as input the gas age distribution responsible for the smoothing of the NEEM record. For this, we chose the age distribution estimated with a gas trapping model at the modern site of Siple Dome which has an accumulation 10.8cm ie yr-1 (Witrant et al., 2012), similar to the accumulation of NEEM during the DO21 period (around 11.3cm ie yr-1 , Rasmussen et al., 2013). In order to test the sensitivity of our results to the deconvolution step, we performed several deconvolutions with age distributions of modern sites ongoing accumulation values above and below the range of potential NEEM accumulations during the DO21 period. The results are presented in Section S2 of the Supplement and indicate that the deconvolution of NEEM is well constrained, and only weakly depends on the specific choice of the age distribution. The effect of the deconvolution is mainly to increase the amplitude of the fast feature at the onset of the DO21, which in turn increases the consistency between the NEEM and Vostok methane records.*"

We will also rewrite the section of the Supplement **SP1L1**:

*"In the main article we use high-resolution methane records as input atmospheric scenarios to study the smoothing of low-accumulation records. However, high-accumulation records can only be used in such a way if the fast variability is well preserved and not affected by smoothing. In the case of NEEM, this appears not to be the case. It can be seen in Figure S2 that it is not possible to find an age distribution to transform the NEEM record into the Vostok DO21 record. In particular, it is not possible to reproduce the Vostok record during the first feature of the DO21 event (1260m depth in the record), as the magnitude of methane peak is not high enough in the NEEM record. It means that in this case, NEEM cannot be directly used as the input atmospheric scenario. This suggests that the NEEM scenario needs to be deconvoluted first, to retrieve the full atmospheric variabilty. To do this, one needs to assume an age distribution for the NEEM methane record. During the DO21 period, the NEEM accumulation has been estimated to be between 10.4 and 13cm ie yr-1, with an average value of 11.3cm ie yr-1 (Rasmussen et al., 2013). We therefore chose to use the age distribution estimated at the modern site of Siple Dome (10.8cm ie yr-1, Witrant et al., 2012). In order to test the sensitivity of the results to the choice of GAD, we also performed the deconvolution with the age distribution of South Pole (8.0cm ie yr-1) and Berkner Island (14.1cm ie yr-1, both accumulations from Witrant et al., 2012). The three deconvolutions produce the atmospheric scenarios displayed as solid lines in Figure S3. When these three scenarios are smoothed by the glacial distribution chosen in Section 3.4.2 of the main article, they result in similar low-accumation records, displayed as dashed lines in Figure S3. We also performed the gas age distribution extraction procedure of Section 3.4.1 of the main article, and found similar gas age distributions (within uncertainty range) using the three deconvolutions as atmospheric references. It indicates that our results weakly depend on the particular choice of GAD for the deconvolution of the NEEM record."*

(1) Could NGRIP be used instead of NEEM? Would the NGRIP record also need to be deconvolved to be consistent with Vostok at DO21?

The NGRIP discrete data do not resolve the first feature of the DO21 as well as NEEM. We tested the use of NGRIP, but even after deconvolution, the NGRIP as atmospheric references result in a small underestimation of the magnitude of the first feature of the DO21.

(2) Can the authors say more about why there is a need to deconvolve the NEEM signal while other reference datasets appear to work without the deconvolution?

Our understanding is that we do not need to deconvolve the WAIS and Fletcher records as these two sites have higher-accumulation of 15 and 20 cm/yr, respectively. Moreover, the DO21 event is more sensitive to smoothing, which also means that it is more likely to be affected in the high-accumulation record.

(3) I can't help but notice there are actually not any new CFA data at 1260 m in Vostok. There is a large data gap between 1260-1261 m or so. So the authors are assuming the height of the CH4 feature there? How did they do this?

The GAD extraction procedure only takes into account the part of the record where CFA data is available. In the case of the DO21, the choice of the GAD is made to match the flanks of the first features, as well as the second large methane increase. We therefore do not have to assume what is the Vostok signal in the gaps. However, the presence of gaps in the data remove constraint for the determination of the GAD. This might lead to larger uncertainty range on the final GAD.

We will add **P15L19**:

*"The GAD extraction procedure is designed to match the CFA record and the convoluted atmospheric reference only where CFA data are available. We thus do not have to extrapolate the CFA data within the gaps of the record. However, the absence of gaps in the records would have reduced the uncertainty of the estimated GAD, as the additional data would have provided more constraint on the GAD estimation."*

Section 3.2 – A table showing which atmospheric reference is paired with which new dataset would be helpful. Additional information such as the age scale of the atmospheric reference dataset, the time interval, the measurement technique used, the resolution of the data, the accumulation rate at the core site, and any treatments to the reference dataset (e.g., spline fit, deconvolution) could also be listed.

These information will be put in the sites' table.

P8Fig2 – Legends on this figure and others would be very helpful. Please label the events in the figures (e.g., DO6, DO7, DO8, etc.). Why are there data gaps in the new CFA records? For this figure and others that follow, plotting the atmospheric reference on depth is very confusing. The depths are obviously not the same in the reference core as for the new data – they are from different cores.

P8Fig3 – See comments about Figure 2 with respect to plotting the reference records on depth.

We will modify the figures of the article to put the gas age scale as the main axis. We will use the secondary top-axis as the low-accumulation ice core depth scale.

We will label the different DO events, as well as the 8.2 ka event, in the figures.

The gap in the records corresponds to sections that could not be measured, either because the ice was unavailable, or because the ice was too broken to be cut into CFA sticks. This will be explained in the methods section.

P9Fig5 – For this figure and for others, the reader must follow multiple cross-references to different captions in order to understand what is being plotted. Consider adding legends to the figures for clarity, or writing full captions for each figure rather than cross-referencing.

We will add legends to the figures.

P9Fig5 - Please explain how one can measure a layer-trapping artifact that is higher in CH4 concentration than any measured value during DO8? I'm referring to the highest light blue value in Figure 5 near 740 m.

After investigation this section appears to correspond two ice cores sticks with internal cracks (that were observed and reported in our measurement campaign log book). Because of these internal cracks, modern air can enter the system and lead to abnormally high methane values. This type of contamination is normally removed from the dataset, but this one was not detected and was let in the final dataset. As this section appears to be contaminated by modern air, we have removed it from the dataset.

Moreover, the section is about one meter long, and therefore cannot correspond to a layering artifact that is typically a few centimeters thick.

P9L2 – This is the first mention of the analytical noise. Consider putting this in the methods section where the authors discuss the analytical technique.

We will put in the description of the CFA "*Using a mixture of de-ionized water and standard gases, the analytical noise of the CFA system has been determined to be about 10ppbv peak-to-peak (Fourteau et al., 2017)*".

P10L5-8 - This is a very fundamental and robust conclusion – that late close-off artifacts are rare relative to early close-off artifacts. What does that mean for the GADs? Is there

The GAD in late closure layers might indeed be affected. As the trapping will be spread over a shorter period of time, we might expect a narrower GAD in late closure layers. However, we are not able to quantify or observe this effect at this point.

**P11L3 – What about Na+ or other ions? How good is the assumption that Ca2+ is a predictor of total ion content?**

The high-resolution Lambert et al. (2012) dataset only includes mineral dust and calcium (which is largely dust-derived), so we cannot test the potential influence of Na+ or other major ions. However, the study of Hörhold et al. (2012) has highlighted the link between calcium variability and deep firn stratitification in several firn cores from Greenland and Antarctica. Fujita et al. (2016) show that calcium is correlated with several ions including Na+, K+, Cl-, and F-. We will modify the text of the article to emphasize that calcium variability can be used as a proxy for density variability, but that there is not necessarily a causal link between the two **P10L18**:
*" We chose to focus on calcium since high resolution data are readily available and since Hörhold et al. (2012) observed a correlation between the calcium variability and the density variability of firn. However, as pointed out by Hörhold et al. (2012) and Fujita et al. (2016), it does not entail that calcium is the ion responsible for the establishment of deep firn stratification. Indeed, calcium is correlated with other ion species that could be the cause for the preferential densification of some firn strata (Fujita et al., 2016)."*

**P11L6 – How many volcanic markers are there common to both cores in the depth intervals relevant here?**

The information will be added in text *"The depth difference between the EDC96 ice core, in which the methane measurements were performed, and the EDC99 ice core, in which the calcium measurements were performed, was taken into account using 18 individual volcanic markers (Parrenin et al., 2012)"*

**P11L10 – Exactly how are the layer trapping artifacts identified? This should be explained more clearly, possibly in the methods section. How can the authors be sure these are not analytical artifacts?**

In this case we identify them manually, by searching for abrupt centimeter-scale methane variations with amplitude of more than 10ppbv (thus larger than analytical noise). For negatively orientated artifacts, we know that they cannot be caused by contamination during measurements, and we can therefore safely assume that they correspond to low methane concentrations in the ice. Positively orientated values can be mistaken with the intrusion of laboratory air in the CFA system. That is why, when an intrusion might occur in the system during measurements, it is recorded and later cleaned from the data. It however remains possible that non-cleaned intrusions remain in the system, and are mistakenly identified as layering artifacts. However, the positions of the positive spikes that we observe are near periods of atmospheric variations, which is consistent with the mechanism of layered gas trapping, and suggests that most of them are layering artifacts and not contaminations.

As explained in the Major Comments Section, we will modify the article **P9L1**:
*"In accordance with Rhodes et al. (2016) and Fourteau et al. (2017), we observe abrupt centimeter-scale variations in the records during periods when the atmospheric CH4 concentration was varying, that we interpret as layering artifacts. We identified such layering artifacts as spikes with widths of a few centimeters and whose concentrations are larger than the analytical noise of ~10ppbv. Despite the effort to clean the record of modern air contaminations, it is possible that some contamination spikes remain and would then be wrongly interpreted as layering artifacts. However, the presence of negatively orientated spikes that cannot be attributed to contamination,*

*and the overall distribution of spikes during periods of atmospheric methane variation confirm that these abrupt variations are mostly due to the mechanism of layered gas trapping."*

P11L20 – Do the DO17 data also show a high number of layer artifacts?
As explained above, we have decided to remove the calcium parameterization from the layering artifacts model because it's use is supported by very limited data. We will explain in the manuscript that simultaneous high resolution measurements of chemical tracers and methane in the same ice core sections are needed to investigate the effect of chemistry on layered gas trapping.

P11L26 – What do the authors mean by the "bulk behavior?" How is that defined?
By bulk behavior we mean the average meter-scale behavior of firn, with centimeter scale variations smoothed out. We will reword the whole text to remove the work bulk.

P12L14 – How sensitive is the model to the assumed densification rate?
A halving of the densification rate results in a doubling of the depth closure anomalies and age anomalies. In this case, the zones of layering artifacts (blue and yellow envelopes in Figure 2 to 6) would stretch further, resulting in layering artifacts being present in parts where they initially were not.
Note that a detailed sensitivity analysis of the layered gas trapping model is provided in the Supplement of Fourteau et al. (2017).

P13L6 – I don't see a "clear underestimation" of the layering artifacts for DO events 7 and 8 in Figure S4. Please clarify what the authors see in the data.
It would be helpful to also plot the reference records in the upper panels of these plots so readers can compare how the data look relative to the better record of atmospheric history.
P13L9 – I don't understand which visual results the authors are judging to be correct here, so the 1.5 factor seems arbitrary.
What we meant is that while the model predicts that artifacts should not be present above 749m (end position of the blue envelope), we observed a few layering artifacts past that point in the EDC99 record. It suggests that some early closure layers might have closure depth anomalies larger than what is predicted by the model. In order to account for this observation we added the enhancing factor, that what empirically set to 1.5. This was done to improve the position of the artifacts predicted y the model.

However, we are aware that this point is not robust as it is only based on the observation of a few layering artifacts. As this enhancing factor only marginally improves the model, we have decided to remove it from the article. We will present the model without this modification, as it is able to adequately predict the position and amplitude of most layering artifacts, even though improvement could still be made.

P14L3 – How is the analytical noise estimated?
During the cleaning procedure, the analytical noise is estimated by computing the Normalized Median Absolute Deviation (NMAD) of the data. This will be added in the text:
*"Then, the analytical noise is estimated using the Normalized Median Absolute Deviation of the data (NMAD, Rousseeuw et al., 2011; Fourteau et al., 2017). The data are clipped above 2.5 times the value of the NMAD, in order to trim a part of the artifacts."*

Please note that we will also provide additional information on how the cleaning procedure operates, notably by pointing out that it works in a recursive way to progressively trim the artifacts. We will add **P27L13**:
*"Hence, to clean the new three methane signals presenting layering artifacts (EDC96, EDC99 and*

*Vostok) we use a recursive cleaning procedure similar to the algorithm described by Fourteau et al. (2017). Briefly, this cleaning algorithm starts by estimating a smooth signal that should represent the measured signal free of layering artifacts and analytical noise. For this purpose, a running median is first computed to remove the layering artifacts while minimizing bias. Then, the signal is smoothed with a binned average, and interpolated back to high resolution using an interpolating spline. Then, the analytical noise is estimated using the Normalized Median Absolute Deviation of the data (NMAD, Rousseeuw et al., 2011; Fourteau et al., 2017). The data are clipped above 2.5 times the value of the NMAD, in order to trim a part of the artifacts. The algorithm is then looped until the signal is determined to be free of layering artifacts. The signal is considered to be free of artifacts when the NMAD (estimation of noise without the layering artifacts) and the standard deviation (estimation of noise with the layering artifacts) are similar (Fourteau et al., 2017). This recursive method produces a progressive removal of the layering artifacts."*

P14L5 – Here (in the Holocene data) I can't help but question if the authors have actually identified real layer trapping artifacts or just the outlier analytical noise. Can they really distinguish? Please provide some justification.

The spikes visible in the middle of the record (~323m) are much larger than the analytical noise, and are systematically oriented toward high-value. They are therefore unlikely to be simply analytical noise. For the rest of the record, where the signal is barely trimmed, we agree that is the removed data are simply analytical noise. We will clarify **P14L5**:

*"This procedure was successfully applied to the Holocene section of EDC99, that only exhibits a few artifacts near the depth 323m. Note that the cleaning algorithm also removes a small part of the analytical noise. However, this small removal of the analytical noise is negligible and does not influence our conclusions."*

P14L11 – Please justify why providing a manually specified artifact-free signal is not circular. If the authors can visually identify the artifacts so straightforwardly, why then is it necessary to run the algorithm at all?

The algorithm is not meant to help us identify the artifacts, but to remove them from CFA data set. Such tasks could be done purely manually, but will be lengthy and cumbersome due to the large amount of data in the CFA signals. We will make this point clearer **P13L26**:

*"As the layering artifacts can be visually distinguished, it would be possible to remove them manually from the data. However, such procedure would be cumbersome due to the large amount of CFA data points. Hence, to clean the new three methane signals presenting layering artifacts (EDC96, EDC99 and Vostok) we use a recursive cleaning procedure similar to the algorithm described by Fourteau et al., 2017)."*

For the case of EDC99 and Vostok, we rely on the algorithm to estimate the analytical noise, and perform the progressive trimming of the data. Our goal is to still have a mostly-automated process to clean the data, and to ensure consistency with the other records. We will add P14L13:

*"The rest of algorithm then proceeds as normal, which allows us to use the automated determination of analytical noise and cleaning of the data. This also ensures a better consistency of cleaning between the different CFA datasets."*

P17L6-10 – This seems like the main point of this paragraph and possibly one of the more important conclusions of the paper. Consider starting the paragraph with this sentence and filling in the details thereafter.

We will modify the text, starting **P16L6** with :

*"Our results suggest that, during the glacial period, East Antarctica ice cores are affected by the same level of smoothing, that can be represented with the same gas age distribution. We produced*

*two new glacial period gas age distributions (EDC96 DO6-9 and Vostok DO21), that are to be added to the previously published GAD obtained for the Vostok site during the DO17 event using the same GAD extraction method (1.3cm ie yr-1 accumulation rate; Fourteau et a 2017). These three age distributions of East Antarctic sites under glacial conditions are displayed in the right panel of Figure 9. The smoothing they induce is represented for the DO6 to 9 and DO21 events in Figure 10. It appears that the two Vostok distributions lead to a similar smoothing. On the other hand the GAD obtained for EDC96 for the DO6-9 events is significantly broader than the Vostok ones and results in a stronger smoothing, especially visible at the onset of the DO21 event. This is surprising as the Vostok DO21 and Dome C DO6-9 records have the same accumulation rate, and should therefore present similar age distributions.*

*However, the uncertainty analysis in Section S5 of the Supplement reveals that the age distribution extracted from the EDC96 record is poorly constrained. This indicates that the smoothing of DO6-9 period is not sensitive to the choice of GAD, and that a large range of GADs results in adequate smoothing for the EDC96 record. On the other hand, the DO21 period is very sensitive to the choice of GAD, and fewer age distributions are able to reproduce the smoothing of the Vostok DO21 record. Our understanding is that because of its shape and its fast atmospheric rate of change (Chappellaz et al., 2013), the first feature of DO21 is sensitive to the choice of GAD, despite a gap in the record, and is therefore a good discriminant between potential age distributions. On the other hand, the step-like features of the DO6-9 record are less sensitive to the choice of GAD, which leads to a less-well constrained extraction of age distributions. Consequently, the distributions obtained for glacial Vostok (DO17 and 21 events) are also suited for the smoothing of DO6 to 9 events in EDC96, as seen in the lower panel of Figure 10. This suggests that the smoothing of the three glacial records are similar, and that the three ice core therefore encloses similar gas age distributions. We thus propose to use a common gas age distribution to represent the smoothing in ice cores with accumulations below 2 cm ie yr-1. The distribution proposed by Fourteau et al. (2017) is a good candidate for this common distribution, as it reproduces well the smoothing in the EDC96 DO6-9 record and the Vostok DO17 and 21 records. Moreover, its shape is a compromise between the two other glacial GADs proposed in this article (right panel of Figure 9)."*

P17Fig10 – I'm still having trouble understanding how the atmospheric references can be plotted on the depth scale of a different ice core. Also consider plotting the atmospheric reference in dashed orange lines to be consistent with previous figures.
We will modify the x-axis and change the color of the atmospheric reference.

P17Fig10 - I would like to see more discussion of why the various GADs estimate the smoothing of DO6-9 similarly well, but they have larger discrepancies for DO21.
The DO21 event is better suited to distinguish between GADs because the smoothing of its first feature is more sensitive to the choice of the GAD. On the other hand, the DO6-9 events are less sensitive to smoothing, and different GADs produce relatively similar smoothed signal.
It will be made clearer in the text (see our comment **P17l6**)

P18Fig11 – Again the reference is plotted on depth, which is very confusing without further explanation. Consider plotting in dashed orange to be consistent, similar to previous comment.
The requested modifications will be made.

P19L2 – How does the skew relate to the firn? What's the physical reason for the skew?
The skewness suggests that most of the trapping occurs at the very bottom of the firn, that is to say that a large part of the porosity rapidly transitions from a open to closed ice. However, we do not

have any independent observations to confirm this phenomenon.

P19L8 – Can the authors explain this conclusion further? Readers may think synchronizing to a high-accumulation record would increase the accuracy of the gas chronology.

Synchronizing to a high-accumulation record indeed increases the accuracy of the gas chronology, because the chronologies of high-accumulation records are better constrained. What we mean is that to take full advantage of the precise chronology of high-accumulation records, one has to perform the synchronization on signals with similar degrees of smoothing (by convolving or deconvolving the signals), in order not to be affected by phase shift effects. Please note that smoothing a high-accumulation record do not deteriorate its chronology.

We will clarify the text **P19L8**:
"*To produce consistent chronologies, the synchronization should be performed on signals with a similar degree of smoothing (either by convolving the higher-accumulation record or deconvolving the lower-accumulation one). Otherwise, the phase shifts should be taken into account in the age uncertainty estimates.*"

P21Fig12 – The artificial depth scale is confusing, can you explain where it comes from?

It is a relative depth scale. As this is a synthetic signal, we cannot produce an absolute depth scale. The figure will be redrawn. We will add in the text **P20L7**:
"*As this is a synthetic signal, we plotted the data on a relative depth scale, whose zero was arbitrarily set in the record.*"

PS1Fig1 - Again the NEEM reference data are plotted on the same depth scale as the Vostok DO21 data, which is confusing.

We will modify the x-axis.

PS1Fig1 - I think a problem with the point the authors are trying to make here is that there is a large data gap at the onset of DO 21. The full magnitude of the CH4 rise in the Vostok DO21 record is not technically resolved

As explained in the answer to the comment (3) of Section 3.2, the fact that the CFA measurements do not resolve the full peak is not problematic, as the choice of the GAD is made to match the rest of the signal.

---

## Author Comment (AC2) · 15 Nov 2019

**Response to Referee #2 – cp-2019-94**

We are thankful to the referee for their constructive comments on the article.
We will improve the manuscript clarity and detail the methods section following the guidelines provided by the referee#1.

Below are comments of the referee in blue, with our corresponding responses in black.

Kévin Fourteau on behalf of all co-authors

P3, Line15: Please write the units out the first time. 2 cm ice equivalent yr-1 (cm ie yr-1). No dot between cm and ie; otherwise it means cm times ice equivalent.
We will modify the units throughout the article, and remove the dots.

Please make it clear early in the manuscript that you prefer to use Antarctic high resolution data for comparison as they are not affected by the pole to pole gradient.
We will add to the text **P7L1**:
*"When possible, we use high-accumulation records from ice cores drilled in Antarctica. Otherwise, in order to compare the high and low-accumulation records we need to estimate the methane inter-hemispheric gradient."*

There are two continuous records from NEEM. Explain why you prefer to take the one you do or explain that it does not matter, does it?
We took the average values of the two instruments used by Chappellaz et al 2013. That being said, the two dataset are similar, and we could have also chosen one or the other.
We will add **P7L8**:
*"Finally, the atmospheric reference used for the Vostok DO21 period is based on the NEEM CFA data published by Chappellaz et al (2013). Chappellaz et al (2013) propose two CFA records obtained with two different spectrometers. For this study, we used the average values of the two instruments"*

I have a feeling on how the model works but the mathematical formulation on page 12 does not make sense. Unit wise that equation is definitely wrong. The model needs to be explained in depth and better before this manuscript is publishable.
Equation1 P12 is homogeneous, both the right and left hand side are expressed in m, the right hand side being kg m-3 / kg m-4. For clarity we will modify the way the unit of the densification rate is expressed in the article, using kg m-3 m-1 rather than kg m-4.

We will also remove the calcium-parametrization because it lacks robutness. The model is therefore now simply the depth-anomaly based version of the Fourteau et al (2017) model.

The manuscript has too many figures where records are also unnecessarily repeated. I suggest fewer graphs. The graphs also lack information on which record the Antarctic data is compared to. Please label Dansgaard-Oeschger events in the manuscript. When sections of the core are compared in the text, it would help to have them labeled in the graph.
We do not understand what repeated figures the referee is referring to. To gain space, we have merged both panels of figure 12.
We will label DO events in the figures, as well as the 8.2 ka event.
We will also add a table summarizing the studied ice core sections and their corresponding atmospheric reference.

Supplemental: S1 The depth scale seems to apply to Vostok not to NEEM. What section of the NEEM core is that? It is quite obvious that there is a gap in the original record that leads to the too much smoothed record. The conclusion about NEEM gas age is not supported in my opinion.

The depth scale is the one of Vostok. The mesured section is the DO21 of the main article. We will modify the figure to use a gas age scale as the main x-axis. We will also add a second axis with the equivalent depth in the Vostok record.

The adjustment of the GAD is done solely where data are available, which means that in the DO21 Vostok case the adjustment is performed using the flanks of the first methane excursion and the large second excursion. The inability to find an age distribution that both matches the second excursion and the flanks of the first one is what leads us to think that the non-deconvoluted NEEM data cannot be used as an atmospheric scenario for the Vostok ice core. The missing data at the peak of the feature cannot explain why it is no possible to find an age distribution that matches the flanks of the feature.

We will clarify this point in the text **P15L19**:

*"The GAD extraction procedure is designed to match the CFA record and the convoluted atmospheric reference only where CFA data are available. We thus do not have to extrapolate the CFA data within the gaps of the record. However, the absence of gaps in the records would have reduced the uncertainty of the estimated GADs, as the additional data would have provided more constraints on the GAD estimation."*

---

## Author Response (AR1)

Dear Professor Fischer,

Please find below the revised versions of the manuscript and the supplement of cp-2019-94. The differences with the first versions are highlighted, with removed text in strikeout red and added text in underlined blue.

Following the comments of the reviewers, here are the main differences with the first version:

- We have added details to describe the CFA method, notably why gaps are present in the final data.
- We have clarified the supplement section explaining why we need to deconvolve the original NEEM record, and we have performed a sensitivity analysis to asses the impact of the deconvolution on the conclusions of the article.
- We have removed the calcium parameterization from the layering artifacts model as it lacked robustness. However, we have kept the discussion on the potential link between chemistry and the presence of layering artifacts (Section 3.3.1). We have clearly stated that this is a preliminary results, that needs to be confirmed with dedicated high-resolution measurements. However, we do not oppose removing this section from the article as well, if requested to improve the clarity of the article.
- We have added details on the algorithm to remove layering artifacts from the CFA, and explained why we do not clean the data purely manually.
- We have detailed the description of the GAD extraction method.
- We have added a more detailed discussion on why the DO21 period is a better discriminant for gas age distributions than the DO6-9 period.

The removal of the calcium parameterization in the layering model should facilitate the overall understanding of the paper. We are aware that different parts of the manuscript are not independent and that these overlaps don't facilitate the reading, but we did not find a simpler overall organization of the manuscript.

Best Regards,
Kévin Fourteau on behalf of all co-authors.

[revised manuscript text omitted]

 The CFA records measured at IGE need to be corrected for the preferential dissolution of methane compared to oxygen and nitrogen in water (Chappellaz et al., 2013; Rhodes et al., 2013). This was done

5  by applying correction factors chosen to match the CFA concentrations with already calibrated data. Such a match is displayed for the DO6 to 9 section in the EDC96 ice core in Figure S1. The discrete measurements used for the correction were obtained by a discrete melt-refreeze method (Loulergue et al., 2008).

[Figure]

**Figure S1.** Illustration of a corrected CFA record (in gray) by matching an already-calibrated record (orange dots, Loulergue et al., 2008).

**S2 Deconvolving the NEEM CFA record**

10 In the main article we use high-resolution methane records as input atmospheric scenarios to study the smoothing of low-accumulation records. However, high-accumulation records can only be used in such a way if the fast variability is well preserved and not affected by smoothing. In the case of NEEM, this appears not to be the case. It can be seen in Figure S2 that it is not possible

to find  an age distribution to transform the NEEM record into the Vostok DO21 record. In particular, it is not possible to reproduce the Vostok record during the first feature of the DO21 event (1260m depth in the record), as the methane peak is not high enough in the NEEM record. It means that in this case, NEEM cannot be directly used as the input atmospheric scenario. This suggests that the NEEM scenario needs to be deconvoluted first, to retrieve the full atmospheric variabilty.

To do this, one needs to assume an age distribution for the NEEM methane record. During the DO21 period, the NEEM accumulation has been estimated to be between 10.4 and $13\,\mathrm{cm\,ie\,yr^{-1}}$, with an average value of $11.3\,\mathrm{cm\,ie\,yr^{-1}}$ (Rasmussen et al., 2013). We therefore chose to use the age distribution estimated at the modern site of Siple Dome ($10.8\,\mathrm{cm\,ie\,yr^{-1}}$, Witrant et al., 2012). In order to test the sensitivity of the results to the choice of GAD, we also performed the deconvolution with the age distribution of South Pole ($8.0\,\mathrm{cm\,ie\,yr^{-1}}$) and Berkner Island ($14.1\,\mathrm{cm\,ie\,yr^{-1}}$, Witrant et al., 2012). The three deconvolutions produce the atmospheric scenarios displayed as solid lines in Figure S3. When these three scenarios are smoothed by the glacial age distribution chosen in Section 3.4.2 of the main article, they result in similar low-accumation records, displayed as dashed lines in Figure S3. We also performed the gas age distribution extraction procedure of Section 3.4.1 of the main article, and found similar gas age distributions (within the uncertainty range) using the three deconvolutions as atmospheric references. It indicates that our results weakly depend on the particular choice of GAD for the deconvolution of the NEEM record.

[Figure]

**Figure S2.** Illustration of the  failure in finding a gas age distribution that smooths the atmospheric scenario (in dashed orange, splined NEEM data without deconvolution) in order to match the Vostok CFA measurements (in light blue). The smoothed version in green underestimates the fast event measured in the ice core around the 1260m depth. The black line represents the original Chappellaz et al. (2013) data, that partially overlaps the spline.

[Figure]

**Figure S3.** The NEEM deconvoluted scenarios and their convolution into Vostok records. For each color, the solid line represents the deconvoluted atmospheric reference and the dashed line represents the atmospheric reference smoothed by an East Antarctica glacial age distribution. Orange, blue and green curves respectively represent the cases with a 8.0, 10.8, and 14.1cm ie yr$^{-1}$ accumulation for the NEEM gas age distribution during the DO21 period. The original NEEM data are also shown in black.

**S3 Zoom over layering artifacts**

A closer look at the layering artifacts of the EDC96 DO6-9 section is shown in Figure S4. This highlights the structure of the layering artifacts, and their widths of a couple of centimeters. They are similar to the artifacts reported in Figure 1  of Fourteau et al. (2017).

[Figure]

**Figure S4.** Zoom over the EDC96 DO6-9 methane record. The blue data are the raw CFA measurements. In black are the data cleaned for layering artifacts with the recursive cleaning procedure presented in Section 3.3.3 of the main article.

**5  S4 Comparison of discrete and continuous methane data over the DO8**

In this section we compare the discrete (Loulergue et al., 2008) and continuous (this study) methane measurements over the DO8 event in the EPICA Dome C record. Fourteau et al. (2017) pointed out that one the of the data  points of Loulergue

et al. (2008) might  correspond to an early closure artifact and may not be climatically relevant. This data point is highlighted with a blue circle in Figure S5. The high-resolution measurements confirm that this data point corresponds to an early closure artifact, the sample selected by Loulergue et al. (2008) being in a zone with a lot layering artifacts with similar methane concentrations.

[revised manuscript text omitted]

---

## Author Response (AR2)

Dear Professor Fischer,

Please find a revised version of the manuscript, taking into account your comments. We have provided specific written responses to some of your comments below. Your remarks are in blue and our responses in black.

We have also corrected some extra typos that we noticed during the re-reading of the manuscript.

Please also find below our written responses, a version of the manuscript showing the differences with the previous submitted version, with removed text in red strikeout and added text in underlined blue. The page and line numbers correspond to the numbering of the previously submitted difference version, on which you added your comments.

You noticed that some references exceeded the normal line width, this is a latex bug occurring specifically when making differences within two files, but not in the modified manuscript (without highlighted differences).

Best Regards,

Kévin Fourteau on behalf of all the co-authors.

P5Fig1: I don't think it really needs the Greenland map, but I have no strong opinion about it We agree that the Greenland map is not fundamental, but we added it following one of the comments of the referee #1.

P6L18: please be more precise. What is the exact nominal resolution of the CFA CH4 system including the dispersion in the system?

We added a reference to the article of Fourteau et al. (2017), where the spatial resolution of the CFA is quantified:

"For a similar set-up, Fourteau et al. (2017) showed that the CFA system is able to resolve variations down to the centimeter-scale (50% of attenuation for sine variations with a wavelength of 2.4cm)."

**P7L3: is it possible that the impurity content (both dissolved or particulate) has an influence on the solubility and or gas separation?**

For impurities to have an impact on the dissolution factor, they would need to impact the thermodynamics of methane, oxygen, or nitrogen dissolution in water. However, we are not aware of any major modification of the solubility of gases in water due to the presence of chemical impurities.

P7L5: What is the sampling frequency (CH valies per second or minute)? This is relevant as the analytical uncertainty should create variance (white noise?) on the time/depth scale of the sampling frequency, while the trapping artifacts should manifest on longer time/age depth scales. This is an important criterion for the detection of trapping artifacts!

The spectrometer acquires 6 absorption spectra per second, that are combined to produce one average concentration measurement per second. This has been added to the text P6L17: *"Spectra are acquired at a rate of 6Hz, and these spectra are later averaged to produce one*

"Spectra are acquired at a rate of 6Hz, and these spectra are later aver concentration value per second."

However, the CFA noise is not a white noise, and the observed noise patterns have some correlation length. This comes from the fact that most of the noise in the CFA system does nos originate from the spectrometer, but rather from the transport and extraction line. Therefore, we cannot use a criterion of length or spatial extension to discriminate between noise patterns and layering artifacts. That is why we use the amplitude as the criterion.

P9L2: You may be aware of the fact that recent results show that some CH4 production is taking place in dusty ice during the extraction process of discrete samples for CH4 analyses. During the CFA CH4 measurements this seems not to be the case. This - I call it - in extractu production can add 5-20 ppb in dusty glacial Greenland ice and implies that the published glacial and stadial IGH of CH4 is overestimated. I don't think you can refer to this in your paper, as the first paper that describes this in extractuc effect is not yet published (Lee et al., Geochimica and Cosmochimica Acta, in press) but you should be aware of this. In your case the absolute level of the CH4 results don't play an important role, only the relative changes after smoothing are relevant. You may want to point this out.

We have added P9L2 that our goal is to have coherent methane variability:

"As the Vostok and NEEM sites are located in different hemispheres, it is necessary to take into account the inter-hemispheric methane gradient between the two sites to make the methane variability of two records as consistent as possible."

**P10Fig2:**

The blue and yellow spikes in the lower panel of the figure correspond to the modeled early and late closure artifacts. We made it clearer in the caption:

"The blue and yellow spikes are the randomly distributed modeled early and late closure artifacts, respectively."

P12L8: in response to referee #2, who was suspecting that you also identify some of the analytical noise as trapping artifacts, you could draw from the fact that artifacts have longer wavelength than the analytical noise in your detection algorithm

As explained in our response to the comment P7L5, the noise of the CFA system is a correlated noise, creating variability patterns at the centimeter-scale. We think that the most robust criterion to detect layering artifacts is that they have a much larger amplitude than the analytical noise.

**P21L8 and P227: on the potential impact of chemistry on the GADs.**

One could indeed imagine that the increased level of impurities during the glacial period results in a faster densification, balancing the decrease in temperature and accumulation. However at this point, we do not have any quantification of the potential effect. While it is certainly worth doing more research on this point, it is beyond our results for this paper.

**Estimation of gas record alteration in very low accumulation ice cores**

Kévin Fourteau1, Patricia Martinerie1, Xavier Faïn1, Alexey A. Ekaykin2, Jérôme Chappellaz1, and Vladimir Lipenkov2

[revised manuscript text omitted]
 ( $\frac{3209 \text{m}}{3209 \text{m}}$  above sea level, coordinates  $74^{\circ}08.310'$  S,  $126^{\circ}09.510'$  E). The local accumulation is  $\frac{3.9 \text{cm} \text{ie} \text{yr}^{-1}}{3.9 \text{cm} \text{ie} \text{yr}^{-1}}$  (Yeung et al., 2019). About  $\frac{80 \text{m}}{80 \text{m}}$  of ice was analyzed for methane, ranging from

25 116m-116m (near the firn-ice transition) to 200m-200m depth. Firn air sampling during the drilling operation was conducted down to 108.3m-108.3m depth. The gas record ranges from about 400 to 3000yrBP-3000yrBP (Before Present, with present

**Figure 1.** Left: Map of Antarctica with the sites of Lock-In, Dome C, Vostok, WAIS Divide, and Fletcher Promontory shown (made with the Quantartica package). Right: Map of Greenland with the site of NEEM shown (using Greenland Ice Mapping Project data; Howat et al., 2014).

defined as 1950 CE in this article).

**Modern Dome C ice core:**

A shallow ice core from Dome C was analyzed for depths ranging from 108 to 178m178m. Similarly to Lock-In, this section 5 corresponds to ages ranging from about 400 to 3000yrBP3000yrBP, but is characterized by a lower accumulation rate of 2.7cm ie yr-1 2.7cm ie yr-1 (Gautier et al., 2016).

**Holocene Dome C ice core:**

10

Ice from the second drilling of the EPICA Dome C ice core (referred to as EDC99 hereafter) was measured from 312 to  $\frac{338m}{338m}$  depth. The gas ages range from 7800 to  $\frac{8900 \text{ yrBP}}{8900 \text{ yrBP}}$ , with an average accumulation of  $\frac{3.1 \text{ em} \text{ ie} \text{ yr}^{-1}}{338m}$

 $3.1 \text{ cm ie yr}^{-1}$  (Bazin et al., 2013; Veres et al., 2013). This period includes the 8.2 ka cold event (Thomas et al., 2007), notably characterized by a sharp decrease in global methane concentrations (Spahni et al., 2003; Ahn et al., 2014).

**DO6-9 Dome C ice core:**

15 A section from the first drilling of the EPICA Dome C ice core (referred to as EDC96 hereafter) was analyzed for depths ranging from 690 to  $\frac{780m780m}{780m}$ . The AICC2012 chronology indicates gas ages covering the period 33000 to  $\frac{41000yrBP41000yrBP}{41000yrBP}$ , and an average accumulation rate of  $\frac{1.5cm ie yr^{-1}}{1.5cm ie yr^{-1}}$  (Bazin et al., 2013; Veres et al., 2013). The CH4 excursion associated with the Dansgaard-Oeschger (DO) events 6 to 9 are included in this gas record (Huber et al., 2006; Chappellaz

**Table 1. Summary of the different ice core sections studied in the article with their associated atmospheric references.**

[revised manuscript text omitted]